# Engineered MED12 mutations drive leiomyoma-like transcriptional and metabolic programs by altering the 3D genome compartmentalization

Kadir Buyukcelebi [1], Xintong Chen[2], Fatih Abdula [1], Hoda Elkafas [1], Alexander James Duval[1], Harun Ozturk[1], Fidan Seker-Polat[1], Qiushi Jin[2], Ping Yin[1], Yue Feng[3], Serdar E. Bulun[1], Jian Jun Wei [1,3], Feng Yue [2] & Mazhar Adli [1] ✉

Nearly 70% of Uterine fibroid (UF) tumors are driven by recurrent *MED12* hotspot mutations. Unfortunately, no cellular models could be generated because the mutant cells have lower fitness in 2D culture conditions. To address this, we employ CRISPR to precisely engineer MED12 Gly44 mutations in UF-relevant myometrial smooth muscle cells. The engineered mutant cells recapitulate several UF-like cellular, transcriptional, and metabolic alterations, including altered Tryptophan/kynurenine metabolism. The aberrant gene expression program in the mutant cells is, in part, driven by a substantial 3D genome compartmentalization switch. At the cellular level, the mutant cells gain enhanced proliferation rates in 3D spheres and form larger lesions in vivo with elevated production of collagen and extracellular matrix deposition. These findings indicate that the engineered cellular model faithfully models key features of UF tumors and provides a platform for the broader scientific community to characterize genomics of recurrent MED12 mutations.

Uterine fibroids (UFs), also known as uterine myoma, leiomyomas, and myoma uteri, are benign monoclonal neoplasms of the myometrium[1]. UFs represent the most common gynecologic tumors among reproductive-age women[1,2]. By the age of 50 years, more than 70% of all women (70% white and >80% black) develop at least one fibroid tumor. UFs are non-malignant and not symptomatic in the majority of cases; however, in 15–30% of cases, they disrupt normal uterine functions, resulting in a wide range of severe health problems, including excessive uterine bleeding, anemia, defective implantation of an embryo, recurrent pregnancy loss, preterm labor and obstruction of labor and may mimic or mask malignant tumors in 15–30% of reproductive-age women[1]. Few medical treatments are available for UFs, and many women opt to undergo a surgical hysterectomy. However, such procedures create significant emotional stress on individual patients and a

substantial financial burden on society. These practices are estimated to cost $5.9–$34.4 billion in the USA alone[3].

Advanced genomic tools, including high-throughput sequencing methods, identified recurrent and largely mutually exclusive genetic alterations in UFs. Notably, nearly 70% of UFs tumors harbor somatic mutations in the *MED12* gene, encoding the Mediator Complex Subunit 12 (MED12)[4]. Furthermore, translocation in the high mobility group AT-hook 2 (*HMGA2*) gene, recurrent loss of fumarate hydratase (*FH*), and deletion of collagen *COL4A5-COL4A6* gene are among other recurrent somatic genetic alterations[5]. Recently, Berta et al. identified inactivating mutations in members of the SRCAP complex, which result in H2A.Z loading defects, as additional genetic defects in UFs[6]. Integrated data analysis reveals mutation-specific distinct driver gene expression programs and biomarkers in UFs[6,7], indicating the importance of

[1]Robert Lurie Comprehensive Cancer Center, Department of Obstetrics and Gynecology, Feinberg School of Medicine at Northwestern University, Chicago, IL, USA. [2]Department of Biochemistry and Molecular Genetics, Feinberg School of Medicine Northwestern University, Chicago, IL, USA. [3]Department of Pathology, Northwestern University Feinberg School of Medicine, Chicago, IL, USA. ✉e-mail: adli@northwestern.edu

studying and revealing the mutation-specific aberrant driver targets in UFs.

The mediator complex is a critical determinant of overall genome organization and gene expression program[8–11]. Early reports established that the mediator complex controls transcription initiation and elongation by bridging the regulatory elements (enhancers) with gene promoters and facilitating the RNA polymerase II transcriptional initiation[11,12]. However, more recent evidence suggests a more complex role of the Mediator complex in overall gene expression and 3D genome organization[13–15]. Therefore, the complex is critical for conveying the information between gene-specific transcription factors and the basal RNA polymerase II transcriptional assembly[8,9]. The mediator is structurally assembled from a set of 26 core subunits in humans arrange into three distinct modules: head, middle, and tail, that bind to Pol II as a "holoenzyme"[16–19]. The proper function of these core modules is regulated by a detachable *CDK8* kinase module[9,19], which consists of MED12, CDK8, Cyclin C, and MED13. MED12-dependent CDK8 activation is a critical regulator of the Mediator complex, as oncogenic MED12 mutations disrupt its allosteric regulation of CDK8[19–21]. MED12 connects Cyclin C and CDK8 to the core Mediator and is required for the kinase activity of the CDK8 submodule[8,22]. Recent findings indicate that the *MED12*-containing CDK8 subcomplex can also function independently of the mediator complex[22]. MED12 is involved in several cancer-related signaling pathways related to nuclear receptors, Wnt, and Sonic Hedgehog[10,23,24].

MED12 consists of 45 exons and is located in the *q13* arm of the X chromosome. Initial exome sequencing revealed that 70% of UFs patients display recurrent *MED12* mutations[4], making it the most frequently altered gene UFs[4,5,25]. This finding has been replicated in several studies of various ethnic groups, reporting *MED12* mutation frequencies in up to 92% of all UFs[25]. Significantly, UFs-associated *MED12* mutations are missense mutations that affect the highly conserved region in exon 2. The prevalence of UFs-linked mutations in *MED12* (60%) is missense mutations that add substitutions at three highly conserved *MED12* amino acids: 50% Gly-44 and 10% (Leu-36 and Gln-43); the other 40% of *MED12* mutations related to UFs correspond to missense mutations at different residues or small in-frame insertions and omissions[4,25]. The recurrent alteration in patient samples and the data from the engineered mouse model suggest that altering the Glycine at the 44 amino acid leads to a gain of function "oncogene" mutation that drives UFs tumorigenesis[26].

UFs tumors are believed to originate mono-clonally from a single mutated smooth muscle cell (SMC). However, the mutated SMC co-exists with non-mutated tumor-associated fibroblasts (TAF) in vivo[27]. Critically, when cultured in vitro, these WT fibroblasts outgrow the mutant cells, resulting in the loss of mutant cells and conversion of the population to WT phenotype[28,29]. This hampered the development of cellular models to deeply characterize *MED12* mutations, understand the molecular pathways downstream of these mutations and develop therapeutic targets to inhibit UFs tumorigenesis. To overcome this formidable challenge, we employ CRISPR-mediated homology-directed repair (HDR) to precisely engineer a Gly-44 mutation (Gly→Asn) in a UFs-relevant myometrial SMC line. Critically, the cellular, molecular, and metabolic profiling of the engineered cells highlights that MED12 Gly-44 mutation recapitulates major cellular and molecular phenotypes of MED12 mutant UFs tumors, including altered proliferation, fibroid-like transcriptional program, and genomic instability, indicating that the engineered cells faithfully model several molecular features of UFs biology. Significantly, our transcriptional and 3D genome mapping and analysis show that *MED12 Gly-44* mutations lead to genome-wide 3D chromatin organization and genome compartmentalization.

## Results

### Engineering MED12 Gly-44 mutation in clonal myometrium smooth muscle cells

Notably, more than 50% of all MED12 exon two mutations are in the 44th codon and mutate Glycine into at least six other amino acids (aa),

indicating the significance of Glycine at this position for proper MED12 function. We set out to use CRISPR to knock in a respective mutation containing a DNA template in exon 2 of the *MED12* gene in immortalized myometrial SMC line that retains the expression of various markers of primary SMC[30]. We utilized a CRISPR-based knock-in and qPCR-based single-cell colony selection strategy[31] (Fig. 1a and Supplementary Fig. 1) to introduce Gly-44→Asn mutation in exon 2 of *MED12* and select single-cell colonies. The knock-in template is designed to disrupt the PAM sequence while altering the amino acid at the 44th codon. We transiently delivered WT Cas9, sgRNA, and a custom-designed knock-in single-stranded DNA (ssDNA) oligo template through nucleofection. We then single-cell-sorted, grew individual nucleofected cells, and performed qPCR-based colony screening on genomic DNA from >400 single-cell colonies (Supplementary Fig. 2). We obtained multiple clonal cell lines that are homozygous and heterozygous for Gly-44 mutations (Fig. 1b, c). We validated the WT and mutant allele frequency using CRISPR-TIDER analysis (Supplementary Fig. 3). Critically, since *MED12* is on the X chromosome, one allele undergoes random epigenetic inactivation. Thus, we performed additional screening from the cDNA of the single-cell clones to identify several clonal cell lines expressing the mutant Gly-44 MED12 in the mRNA as validated by cDNA sequencing (Fig. 1c). Western blot analysis shows that the Gly-44 mutation does not alter MED12 protein stability (Fig. 1d).

### MED12 Gly-44 mutation recapitulates UFs-specific proliferation defects

A formidable challenge in establishing an in vitro model of fibroid tumors has been the rapid disappearance of cells carrying *MED12* mutations when the fibroid tumor cells are cultured in vitro[28,29]. This is believed to be due to a reduced proliferative capacity in 2D culture conditions in vitro. To test whether such a slower proliferation defect can cause the disappearance of mutant cells from the population, we initially analyzed the mixed population of cells right after the CRISPR knock-in. Our targeted sequencing and CRISPR-TIDER analysis indicated that the mixed population contained ~36% WT allele, 7% knock-in allele (Gly-44 mutant), and 57% indel alleles (likely *MED12* KO) after initial CRISPR editing. We then continuously cultured this mixed population and performed targeted sequencing at exon 2 of MED12 at the 4th, 6th, 7th, and 9th weeks after the initial gene editing. Significantly, we observed that the mixed population reached ~100% WT allele while gradually losing both the Gly-44 mutant and the indel alleles (Fig. 1e), supporting the observed phenotype of *MED12* mutant primary SMC cells[28,29]. To quantify the proliferation defects in these cells more precisely, we studied clonally expanded pure populations (from single cells) of MED12 WT, Gly-44 mutant, or KO SMC cells. We used Incucyte live cell imaging platform to robustly detect cell proliferation defects by monitoring individual nuclei counts over several days. Critically, the Gly-44 mutant cells are less proliferative than WT cells but better than the KO cells (Fig. 1f). The time course experiment indicates that the WT cells double in ~23 h, whereas the MED12 Gly-44 mutant cells double every ~32 h in 2D adherent conditions. Although interesting, these findings raised the question of how such a mutation drives UFs tumorigenesis if it leads to reduced cell proliferation. We, therefore, tested whether the reduced proliferation is due to the restrictive 2D culture conditions.

### The MED12 Gly44 mutation increased cell-autonomous and non-autonomous proliferation capacity

MED12 Gly-44 mutations are causal in inducing higher cell proliferation and, eventually, fibroid tumor formation in humans and mice[4,32]. To understand whether the reduced proliferation phenotype is an artifact of 2D culture conditions, we cultured the WT and the engineered mutant cells in 3D spheroid conditions. Notably, the MED12 mutants formed significantly larger spheroids than WT cells (Fig. 1g, h),

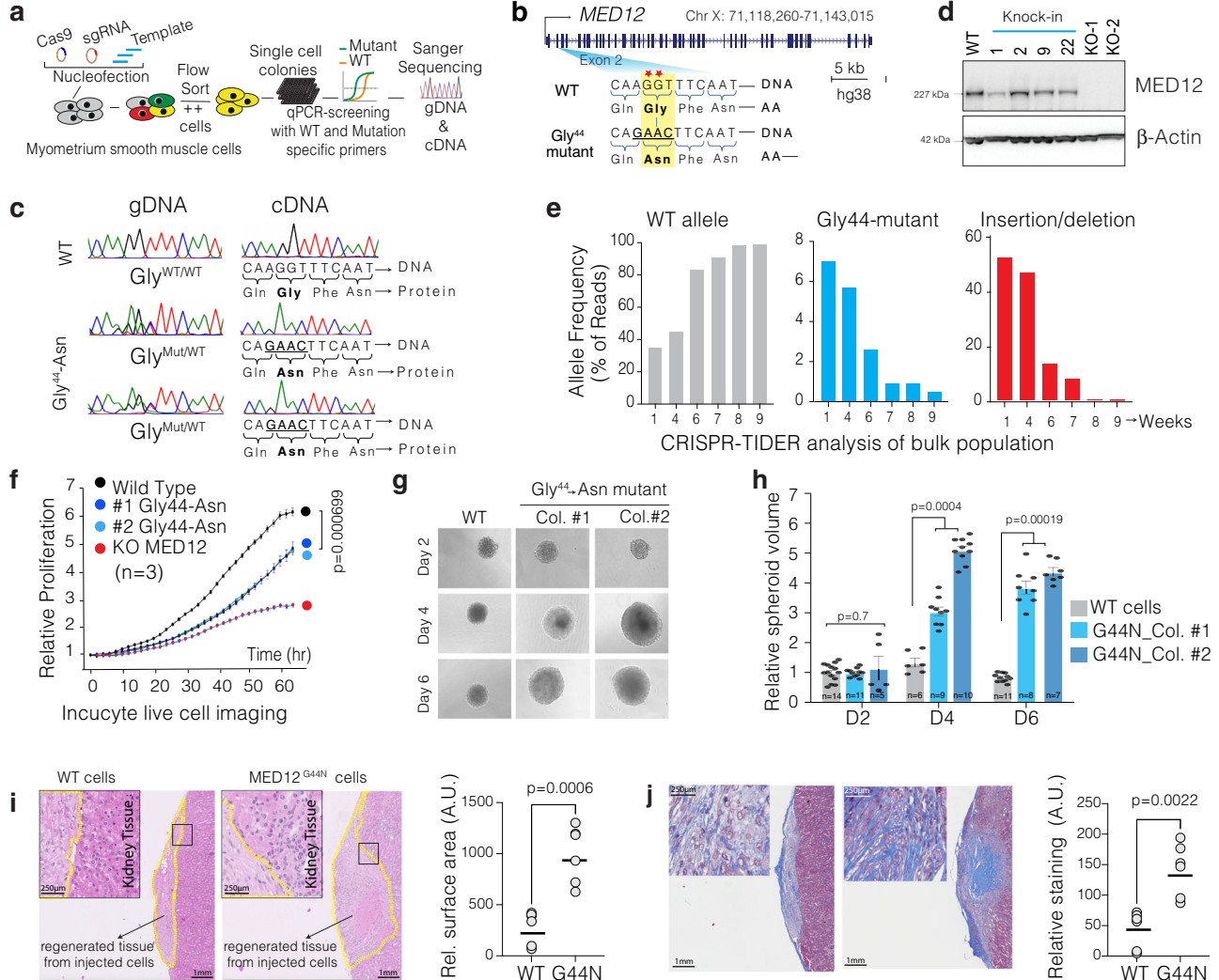

**Fig. 1 | CRISPR-engineered recurrent MED12 Gly-44 mutation results in aberrant cell proliferation. a** Schematics shows the strategy used in this study to CRISPR engineer a UFs-relevant specific MED12 mutation in an immortalized smooth muscle cell. **b** Schematics of the MED12 gene and most frequently observed mutations (red asterisk) are shown at the exon two regions. The WT and the homology-directed repair (HDR) DNA template are shown for the intended MED12 Gly-44→Asn mutation. **c** Chromatograms show the Sanger sequencing results from genomic DNA and cDNA from WT cells and multiple CRISPR-engineered MED12 Gly-44 clones. **d** Western blot showing MED12 and b-actin protein levels in control, CRISPR-engineered cells. **e** CRISPR-TIDE analysis shows the frequency of DNA sequencing reads with WT sequence, homology-directed repaired (MED12 Gly-44 mutant), and indels assessed over several weeks from a population of cells. **f** Incucyte live cell imaging results show relative proliferation rates of control WT and MED12 Gly-44 mutant cells in 2D culture conditions. Results are average of three biological triplicates imaged at 6 different viewpoints. **g** Representative images of 3D spheres of WT and MED12 Gly-44 mutant cells in 3D culture conditions. **h** The bar plot shows relative spheroid volumes from WT and MED12 Gly-44 mutant cells. Results are from one of the biological triplicates and each dot represents a quantified sphere. **i** The H&E images show representative lesions from WT and MED12 Gly-44 mutant cells grown in kidney capsules in vivo. The dot plot quantifies the lesion surface area at the injected site in mice (n = 6). **j** The Masson's trichome staining, showing collagen deposits as blue, indicates overall ECM deposition and collagen production in WT and *MED12* mutant tumors. The dot plot quantifies overall trichrome staining intensity normalized to tumor surface area in different mice (n = 6). Error bars in all figures indicate the standard error of the mean. For statistical analysis in **f**, **h**, **I**, **j**, a two-sided unpaired *t* test has been used.

indicating the significance of culture conditions in studying *MED12* mutations. On average, we observed that MED12 mutant spheres grew 2–4 times larger than the WT colonies. At the same time, the KO cells did not form any spheres, indicating that the engineered MED12 Gly-44 mutation is UFs-relevant and is a gain-of-function mutation.

Notably, the MED12 mutant UFs are composed of mutant SMCs and non-mutant tumor-associated fibroblasts and stromal cells at nearly equal rates[27], suggesting that the mutant cells cause the proliferation of non-mutant cells in a cell non-autonomous fashion. To test whether the engineered mutant cells will also increase the proliferation in non-mutant cells, we performed co-cultured experiments with fluorescently labeled (mCherry) non-mutant cells. We quantified the rate of proliferation in fluorescently labeled non-mutant cells. In line with the data in Fig. 1g, we observed larger spheroids when mutant cells were cultured with non-mutant. We dissociated the spheroids to reveal whether the larger spheroid formation is partly due to the increased proliferation of non-mutant cells. We counted the number of mCherry (+) cells from each condition (Supplementary Fig. 4a). In line with this, we found that the non-mutant cells proliferate at significantly higher levels (p > 0.001, t test) in conditioned media (20%) from MED12 mutant cells compared to media alone or condition media (20%) from non-mutant cells (Supplementary Fig. 4b). These findings indicated that mutant cells enhance the proliferation capacity of the non-mutant cells in the microenvironment.

These results led us to investigate further the proliferation and potential tumor formation capacity of these cells in vivo using a kidney capsule xenograft model. Notably, the *MED12* mutant cells grew a solid lesion, while wild-type cells remained an indiscernible thin layer on the

surface of the kidney (Fig. 1i). Since extracellular matrix (ECM) deposition is a hallmark of UFs, we measured the overall composition of ECM and collagen by performing Masson's trichrome staining, highlighting the overall connective tissues, mainly collagen, in tissue sections[33]. Notably, the tumor-like lesions from MED12 mutant cells contained significantly higher levels of staining ($p < 0.001$, $t$ test) compared to regenerated tissue from WT cells, indicating substantially more ECM and collagen production in the lesions from MED12 mutant cells (Fig. 1j).

### The engineered MED12 Gly-44 mutation alters the global metabolism in smooth muscle cells

The altered proliferation rates are driven by overall cellular metabolic and transcriptional reprogramming. Uterine fibroid tumors have distinct metabolic and bioenergetic needs. For example, uterine fibroid tumors are known to be depleted of Tryptophan but replete with Kynurenine, a product of Tryptophan metabolism[34]. We, therefore, set out to investigate global metabolite levels in WT and mutant cells by performing Liquid Chromatography and Mass Spectrometry (LC-MS) to acquire signals from a broad spectrum of water-soluble metabolites (250 plus targets). The principal component analysis (PCA) of overall metabolite contents (Supplementary Data 1) indicates that the WT and MED12 mutant cells have distinct metabolic states as each formed distinct clusters based on the top two principal components (Fig. 2a), indicating the significant metabolic differences between these two cells. Specifically, MED12 mutation leads to the downregulation of 14 metabolites

while upregulating, a larger number, of 21 metabolites (Fig. 2b, c). Critically, increased 5-HIAA, Kynurenine, thiamine, N-carbomyl-L-aspartate, and reduced Tryptophan, mevalonic acid, N-acetyl aspartic acid, glyceraldehyde were the top differentially regulated metabolites. Tryptophan is an essential amino acid that the body needs to acquire from the diet and is metabolized into Kynurenine. Our detailed quantifications show that while Tryptophan is significantly ($p < 0.01$, $t$ test) depleted in *MED12* mutant cells, Kynurenine levels are abnormally elevated ($p < 0.01$, $t$ test) (Fig. 2d). In line with this, Tryptophan metabolism was the top enriched metabolic term when all differentially regulated metabolites were analyzed together (Fig. 2e). Notably, our group previously reported that the reduced Tryptophan levels in MED12 mutant uterine fibroids are driven by increased levels and activity of an enzyme called *Tryptophan 2,3-Dioxygenase-2* (TDO2) that converts Tryptophan into Kynurenine[35]. We, therefore, tested whether our engineered cells have elevated levels of TDO2 protein. Critically, the western blot analysis shows that the mutant cells have markedly increased levels of TDO2, whose levels are not detectable in WT and the *MED12* KO cells (Fig. 2f). These findings highlight that the engineered Gly-44 mutation recapitulates known metabolic reprogramming in primary UFs tumors[34,35].

### The MED12 Gly-44 mutant cells have distinct transcriptional states

We next assessed overall transcriptional reprogramming downstream of Med12 Gly-44 mutation by performing RNA-Seq in two separate

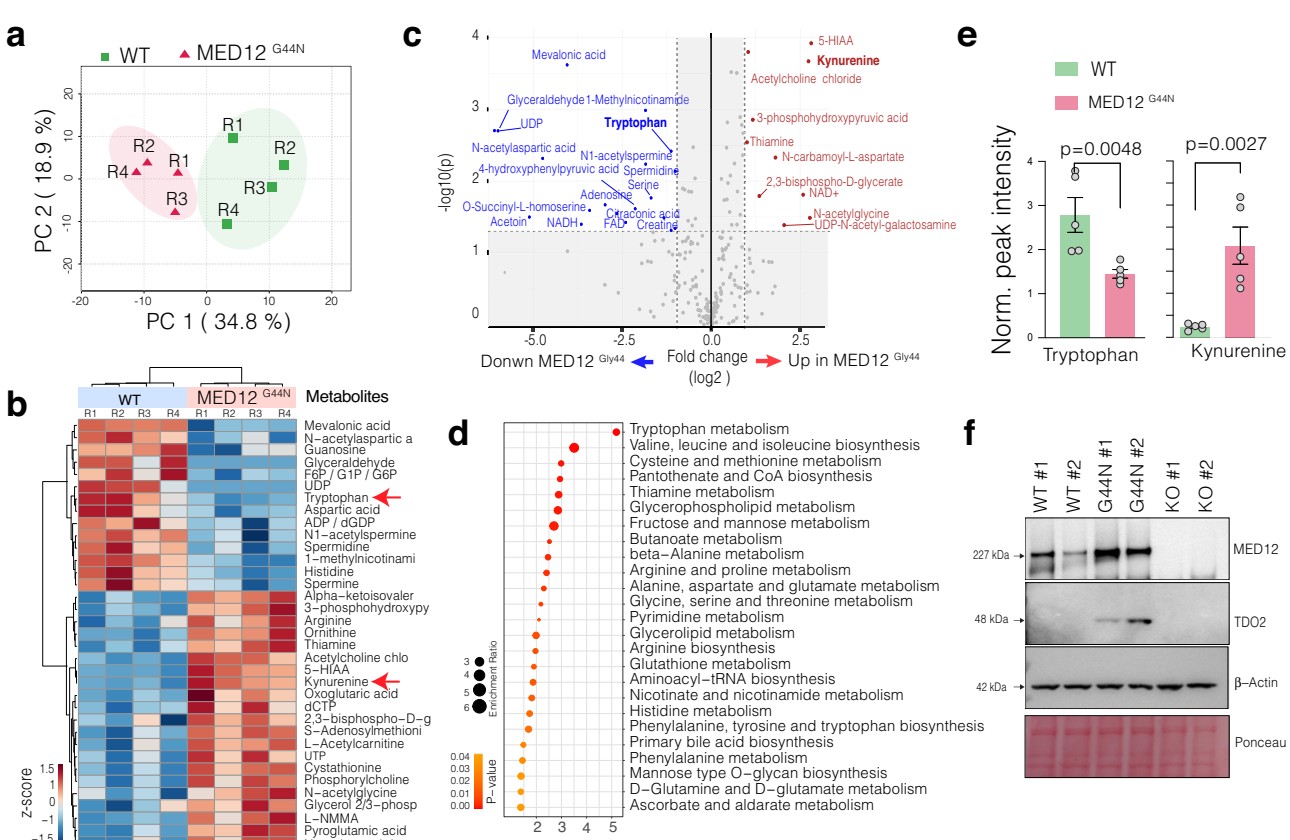

**Fig. 2 | MED12 Gly-44 mutations result in fibroid-relevant metabolic reprogramming. a** Dimension reduction shows the top two principal components that capture 34.8% and 18.9% variability in the global metabolic differences in the indicated samples. **b** The heatmap shows the top enriched and depleted metabolites in WT and MED12 Gly-44 mutant cells. Each raw represents a biological replicate ($n = 4$ total). **c** The scatterplot shows statistical significance (−log10 $p$ value) versus the magnitude of change (−log 2-fold change) in metabolites of MED12 Gly-44 cells vs. WT cells. **d** The bar plots show the normalized intensity of LC-MS peaks

for Tryptophan and Kynurenine metabolites in indicated cells ($n = 5$ replicates). Error bars indicate the standard error of the mean. For statistical tests, a two-sided unpaired $t$ test was used. **e** Dot plots show the $p$ value and enrichment levels of top metabolic terms for the differentially regulated metabolites between WT and MED12 mutant cells. **f** Western blots show MED12 and TDO2 protein levels, and B-actin and Ponceau S staining is shown for loading control. Whole membrane blots are shown in Supplementary Fig. 12.

MED12 Gly-44 mutant clones, the WT and *MED12* KO cells, in triplicate. Notably, >90% of the genes (2124/2359) between WT and mutant cells were consistent across the mutant clones, indicating the robustness of *MED12* mutation-driven transcriptional changes. Less than 5% (104 or 131/2359) of differentially expressed genes were clone-specific, indicating the robustness of *MED12* mutations and minimal clonal heterogeneity compared to WT clones. Differential expression analysis showed that *MED12* mutation drives the differential expression of ~2000 genes, consistent in both *MED12* mutant clones (987 upregulated and 1137 downregulated, (*p*-adj<0.05) (Fig. 3a). Most critically, the MED12 Gly-44 mutations resulted in differential expression of a distinct set of genes compared to *MED12* knock-out cells (Supplementary Fig. 5), supporting the overall hypothesis that the UFs-associated MED12 mutations are not loss of function, but a gain of function mutations.

### The aberrant transcriptional program of MED12 Gly-44 mutant cells is reminiscent of fibroid tumors

To assess whether these differentially expressed genes are comparable and relevant to fibroid tumors, we analyzed them with the recent gene expression program of normal myometrium (*n* = 15) and *MED12* mutation harboring fibroid tumors (*n* = 15) as previously reported by

Moyo et al.[36]. We detected ~5500 differentially expressed genes (*p*-adj<0.01) between normal myometrium and UFs samples (Fig. 3b). Notably, the gene set enrichment analysis (GSEA) highlighted several hallmarks shared among the mutant cells and primary fibroid tumors. For example, hallmarks of cell cycle-related genes, MYC and E2F target genes, and DNA repair genes are substantially upregulated in the mutant cells and primary UFs.

On the other hand, protein secretion and heme metabolism genes are among the most downregulated genes. (Fig. 3c). In line with GSEA, the gene ontology (GO) analyses on differentially expressed genes in Gly-44 mutant cells and fibroid tumors identified a common set of biological processes between these two samples. For example, the upregulated genes in Gly-44 mutant and fibroid tumors are enriched for cell cycle and DNA replication-related gene ontology (GO) terms (Supplementary Fig. 6). Conversely, the downregulated genes in our *MED12* mutant cells and primary fibroids included a group of cell adhesion and extracellular matrix (ECM) reorganization genes, indicating that Gly-44 mutation also recapitulates the abnormal ECM feature of human UFs[37,38].

We next assessed the epigenome of these cells to see whether *MED12* mutation leads to aberrant gene expression changes through the altered epigenome. Since *MED12* is a critical regulator that

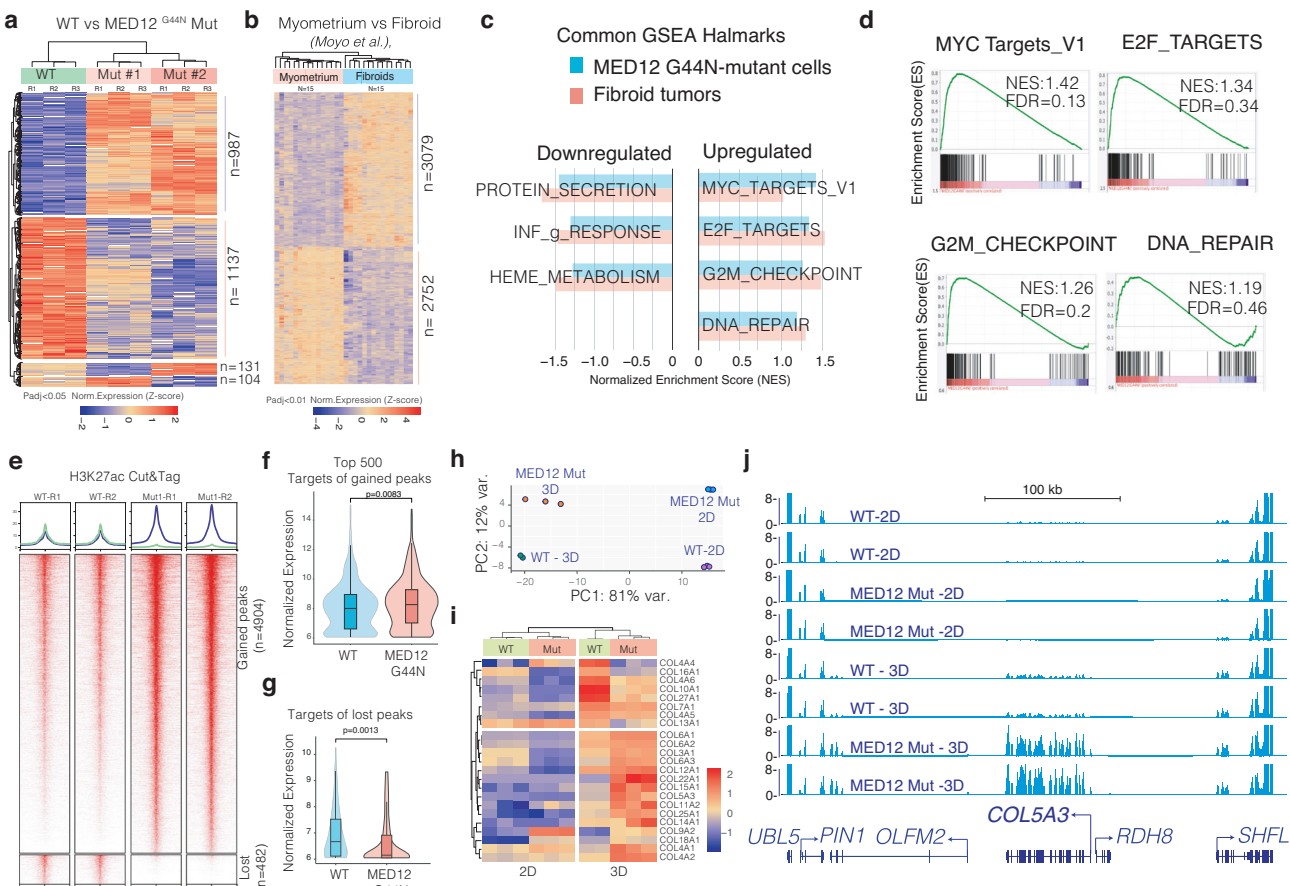

**Fig. 3 | MED12 Gly-44 mutations cause a fibroid-like transcriptional program that is further enhanced in 3D conditions. a**, **b** Heatmaps show differentially expressed genes in engineered MED12 mutant cells (**a**, 3 biological replicates) and primary fibroid tumors (**b**, *n* = 15 sample). Gene names are presented in Supplementary Data 2. **c** The horizontal bar plot shows common hallmarks from Gene Set Enrichment Analysis (GSEA) in engineered *MED12* mutant cells and fibroid tumors. **d** GSEA plots show enriched hallmarks in *MED12* mutant vs. WT cells. **e** Heatmaps show regulatory genomic regions differentially marked by the H3K27ac mark (Cut&Tag signal intensity). **f**, **g** Violin plots and box plots show expression levels of gene targets (<10 kb away from the peaks) of gained (**f**) and lost peaks (**g**) in *MED12*

mutant cells. The Wilcoxon signed-rank test was used for *p* value calculations. In the boxplot, the center line represents the median, the box contains the interquartile range, and the whiskers extend to the 5th and 95th percentiles. **h** The dot plot shows the top two principal components that explain the largest variation in gene expression programs of indicated cell types under two different culture conditions. **i** The heatmap shows differentially regulated collagen genes in WT and *MED12* mutant cells in 2D and 3D conditions. Each raw represents a biological replicate. **j** The RNA-Seq tracks show the gene expression level of *COL5A3* and neighboring genes in WT and *MED12* mutant cells under 2D and 3D conditions.

mediates promoter-enhancer interaction, we acquired a Genome-wide map of Histone H3 Lysine 27 acetylation (H3K27ac), which is associated with enhanced transcriptional activation of histone modification which marks active enhancers and promoters[39,40] by Cut&Tag[41]. Differential peak analysis identified that 4904 peaks gained the H3K27ac mark. At the same time, 482 of the regulatory elements nearly lost all H3K27ac signal (Fig. 3e), indicating that MED12 Gly-44 mutation results in increased genomic activity at most regulatory elements. To identify whether this change in the H3K27ac chromatin state had a corresponding change in target gene expression, we analyzed the expression change in their target genes mapped by their genomic proximity to the H3K27ac peak (<10 kb). Notably, the targets of gained peaks in *MED12* mutant cells had a significantly higher expression in these cells than in WT cells (Fig. 3f). Conversely, the gene targets of lost peaks in *MED12* mutants dramatically reduced their expression in these cells (Fig. 3g). Although it is difficult to pinpoint the causality here, these findings indicated that *MED12* mutations induced gene expression changes partly due to reprogrammed epigenome, at least of the H3K27ac chromatin states.

## MED12 mutations lead to enhanced expression of collagen genes in 3D culture conditions

Fibroid tumors are known to have an aberrantly remodeled extracellular matrix and significantly higher production of collagen[27,37,42]. Recent transcriptome and epigenome profiling by Moyo et al. also highlighted substantially higher expression of collagen genes in fibroid tumors[36]. Surprisingly, we did not find higher expression of collagen genes in our 2D cultured cells. We, therefore, wondered whether this is due to culture conditions. We thus obtained RNA-seq expression profiles of these cells cultured in 3D sphere conditions. Significantly, we found that culture conditions (2D vs. 3D) dramatically impacted gene expression programs. Indeed, >80% of all gene expression variations could be explained by culture conditions, whereas *MED12* mutations induced gene expression alterations contributed to 12% of variations (Fig. 3h). More importantly, under 3D culture conditions, we observed significantly higher expression of collagen and ECM genes in both WT and mutant cells. However, mutant cells had significantly higher expression of the 45 collagen genes (Fig. 3i). Of the 20 that were expressed at higher levels in 3D conditions, and 15 had significantly higher expression in *MED12* mutant cells (Fig. 3i), as exemplified in the RNA-Seq tracks for *COL5A3* gene loci (Fig. 3j). Notably, we observed significant number of differentially upregulated or downregulated genes that are common between the engineered MED12 Gly-44 mutant cells vs. primary fibroids, regardless of culture conditions (Supplementary Fig. 7a, b). Interestingly, we observed substantially higher number of commonly downregulated genes (>2.5-fold increase in Obs vs. Exp) between 3D culture conditions and primary fibroids, indicating the impact of *MED12* mutations is further enhanced in 3D conditions.

## The MED12 Gly-44 mutation alters DNA synthesis and renders cells sensitive to DNA-damaging agents

The above data and cell proliferation rates suggested that MED12 Gly-44 mutation leads to abnormal cell cycle, DNA replication, and repair. We, therefore, tested whether MED12 mutant cells have aberrant cell cycles by analyzing the 5-ethynyl-2′-deoxyuridine (EdU) incorporation and DNA content levels. Notably, the *MED12* mutant cells have a significantly higher percentage of cells in the S-phase ($p < 0.035$), indicating an abnormal rate of DNA synthesis and progression of the DNA replication fork (Fig. 4a and Supplementary Fig. 8). These findings support a recent report that *MED12* mutant UFs samples have increased replication stress due to abnormal progression of the replication fork and increased R-loop formation[43]. Notably, if unresolved, abnormal progression or stalling of replication may lead to structural genomic alterations or excessive DNA damage through replication fork collapse[44,45]. Indeed, complex chromosomal

rearrangements are frequently observed in uterine fibroids[5,46], and *MED12* mutations have been associated with structural genomic variations and genomic instability in mice[32].

The above findings led us to test whether the mutant cells have increased DNA damage at the basal level and are differentially sensitive to DNA-damaging agents such as Carboplatin. To this end, we measured γH2AX, a phosphorylated form of histone H2AX that is deposited at the DNA double-strand break site and can be sued as a proxy for DNA damage/repair activity. Notably, in line with recent findings[43], we did not see a significant increase in total γH2AX at the basal level by western blot (Fig. 4b). However, the immunofluorescence staining shows a detectable difference between overall γH2AX signal intensity, indicating a potential DNA damage/repair activity difference between these two cells (Fig. 4c). Interestingly, we saw significantly more DNA damage accumulation in *MED12* mutant cells in response to Carboplatin in both *MED12* mutant cell lines by western blot and IF (Fig. 4c and Supplementary Fig. 9). The *MED12* mutant Fibroid tumors are known to have higher abnormal progression of the replication fork and increased R-loop formation[43]. Therefore, we tested whether primary fibroid tumors have higher DNA damage/repair activity than normal myometrium. We performed an immunohistochemistry analysis of γH2AX on a well-annotated tissue microarray containing normal myometrium and *MED12* mutant fibroid tumors ($n = 10$). To quantify levels if γH2AX in a robust and unbiased way, we trained a machine learning algorithm to assess γH2AX levels at a single cell level (methods). *MED12* mutant fibroid tumors have detectable and substantially higher overall γH2AX levels, indicating higher DNA damage and repair activity than normal myometrium (Fig. 4d, e and Supplementary Fig. 10). Notably, the γH2AX levels within the same fibroid tumors across smooth muscle cells vs. the stromal cells were comparable (Fig. 4f).

The above findings led us to investigate whether the *MED12* mutant cells that display higher activity of the DNA repair gene, potentially due to a basal level DNA damage, would be selectively more vulnerable to additional DNA damage. We, therefore, performed long-term live-cell imaging to assess relative apoptosis rates (Caspase 3/7 staining, Biotum) over four days. We observed significant apoptotic cell death selectively in *MED12* mutant cells compared to WT cells (Fig. 4g), indicating a potential therapeutically exploitable vulnerability in these mutant cells.

## The MED12 Gly-44 mutations alter genome-wide 3D chromatin organization and genome compartmentalization

The mediator complex is a critical determinant of the overall gene expression program[8–11]. It has also been suggested to play a critical role in genome organization by bridging the regulatory elements (enhancers) with gene promoters and facilitating the RNA polymerase II transcriptional initiation[11,12]. We, therefore, studied whether the abnormal transcriptional program downstream of *MED12* mutations is due to altered 3D genome organization. Depending on the organization scales, the nuclear genome can be categorized into at least three layers in 3D space[47,48]. Globally, the chromosomal DNA is organized into two distinct compartments: A and B. The A compartment is generally associated with gene transcription and active histone modification marks such as H3K27ac and H3K4me3, while the B compartment mainly comprises heterochromatin. At a finer scale (usually ~ hundreds of Kb in size), the mammalian genome is organized as topologically associated domains (TAD), whose boundary can prevent erroneous interactions between the enhancer and wrong target genes. At the finest scale, when a Hi-C library is sequenced deep enough, the chromatin loops that reveal high-resolution promoter-enhancer interactions can be identified. Each layer of genome organization has been linked with proper gene regulation and human diseases[47–50]. To study how MED12 Gly-44 mutation alters the 3D genome organization, we employed high-resolution Hi-C technology and obtained 804,255,654

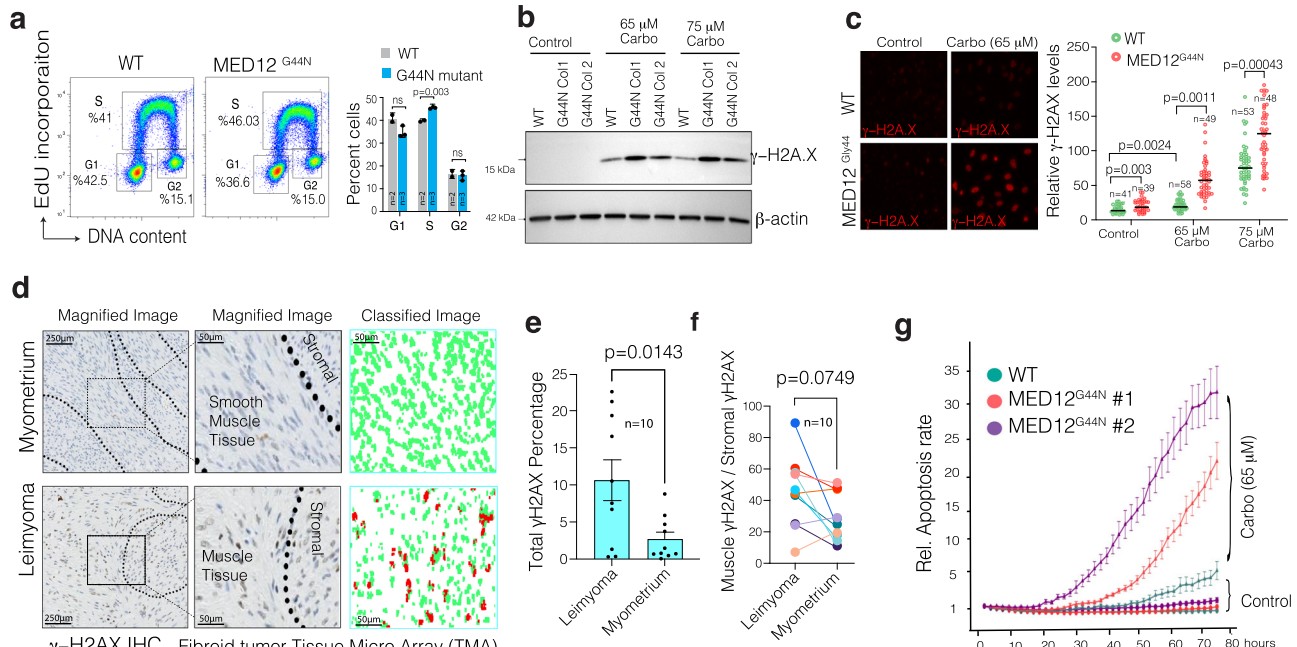

**Fig. 4 | MED12 Gly-44 mutations lead to abnormal DNA repair activity and render cells sensitive to DNA-damaging chemotherapeutic agents. a** Flow cytometry profiles show a representative image of triplicate data showing the rate of EdU incorporation and DNA content analysis in WT and *MED12* mutant cells. **b** Western blots show γH2AX and β–Actin protein levels in WT and mutant cells. Whole membrane blots are shown in Supplementary Fig. 12. **c** The images (quantified in dot plots) show relative immunofluorescence of γH2AX signal intensity in control and Carboplatin-treated WT and *MED12* mutant cells. **d** Immunohistochemistry images show γH2AX staining in normal and MED12 mutant fibroid tumor tissue microarray. The staining intensity in individual cells was assessed by machine learning-assisted segmentation and quantification (see methods). **e, f** The dot plots show overall (**e**) and smooth muscle-specific (**f**) γH2AX IHC signal intensity across ten distinct UFs tissue specimens. **g** The Incucyte live-cell imaging results show relative rates of apoptosis (The Incucyte® Caspase-3/7 dye) WT and MED12 mutant cells measured over more than 3 days in control and Carboplatin-rated cells. For statistical analysis in **a, c, e, f**, a two-sided unpaired *t* test has been used. Error bars indicate the standard error of the mean.

chromatin contact pairs in WT and 717,644,659 contact pairs in MED12 mutant cells (Supplementary Fig. 11a).

We observed a striking difference in Hi-C maps between WT and mutant cells. Compared with WT cells, the *MED12* mutant cell Hi-C map showed much more pronounced plaided or checkboard patterns (Fig. 5a), suggesting the global change in interactions involved with the A/B compartment. We observed significant changes in the compartment state annotations between the two cell types, as exemplified by the eigenvectors track above the Hi-C maps (Fig. 5a). Globally, we found that 7.04% of B compartments switched to A compartment, while 9.45% of B compartments switched to A compartment (Fig. 5b). Integrating gene expression data with 3D genome organization shows that these changes in genomic compartments alter gene expression activity. More specifically, the genes in the compartments that switched from inactive B to active A compartments were significantly upregulated. In contrast, the genes in the A-to-B switching regions were downregulated (Fig. 5c). For example, the *ABCB5* gene, which was in the B compartment and silenced in WT cells, was in a B-to-A compartment region and highly expressed in *MED12* mutant cells (Fig. 5d).

In addition to the compartmentalization switch, we observed enhanced 3D chromatin interactions between the compartments of the same types but reduced inter-compartment interactions. As demonstrated in global average contact frequencies across all compartments in the genome (Fig. 5e), *MED12* mutant cells have significantly higher intra-compartments (A-to-A or B-to-B) interaction frequencies (Fig. 5f). Interestingly, the global compartment strength is primarily driven by enhanced A-to-A interactions and reduced A-to-B interactions, as B-to-B compartment interactions are comparable in WT and mutant cells (Fig. 5f). Finally, we identified 23,487 chromatin loops in WT cells and a comparable number of 23,261 loops in *MED12* mutant cells, using a machine-learning software that we recently

developed[51]. Of these, ~20,000 loops were common, and ~3000 were cell-type-specific chromatin loops (Supplementary Fig. 11b, c). These findings indicate that the differential gene expression changes downstream of *MED12* mutation are partly due to altered 3D genome organization.

Interestingly, a recent report suggested that the MED12/CDK module of the mediator complex controls the global gene expression program, in part, by limiting the formation of dense heterochromatin domains and 3D chromatin compartmentalization[15]. Specifically, Haarhuis et al. observed that MED12 loss enhances H3K9me3-marked heterochromatin domain formations and genome compartmentalization[15]. Notably, the enhanced genome compartmentalization we observe in MED12 Gly-44 mutant cells resembles the findings from MED12-depleted cells. However, while the loss of *MED12* resulted in increased heterochromatin domains and reduced the "open" chromatin regions marked with H3K4me[15], our results show that MED12 Gly-44 mutations resulted in significantly higher open chromatin regions (marked by H3K27ac, Fig. 3e). Furthermore, we find distinct gene expression changes due to MED12 Gly-44 mutation vs. a loss of MED12 (Supplementary Fig. 5). These findings indicate that although Hi-C chromatin compartmentalization in MED12 Gly-44 mutant cells and *MED12* KO cells have some resemblance, MED12 Gly44 mutations result in aberrant expression of a distinct set of genes and altered chromatin accessibility compared to WT or MED12 depleted cells.

## Discussion

Recurrent somatic *MED12* mutations drive fibroid tumors in 70% of cases. Unfortunately, the *MED12* mutant cells from these lesions could not be isolated and maintained in culture conditions[52,53]. This formidable challenge limited the ability to create a tractable genetic model

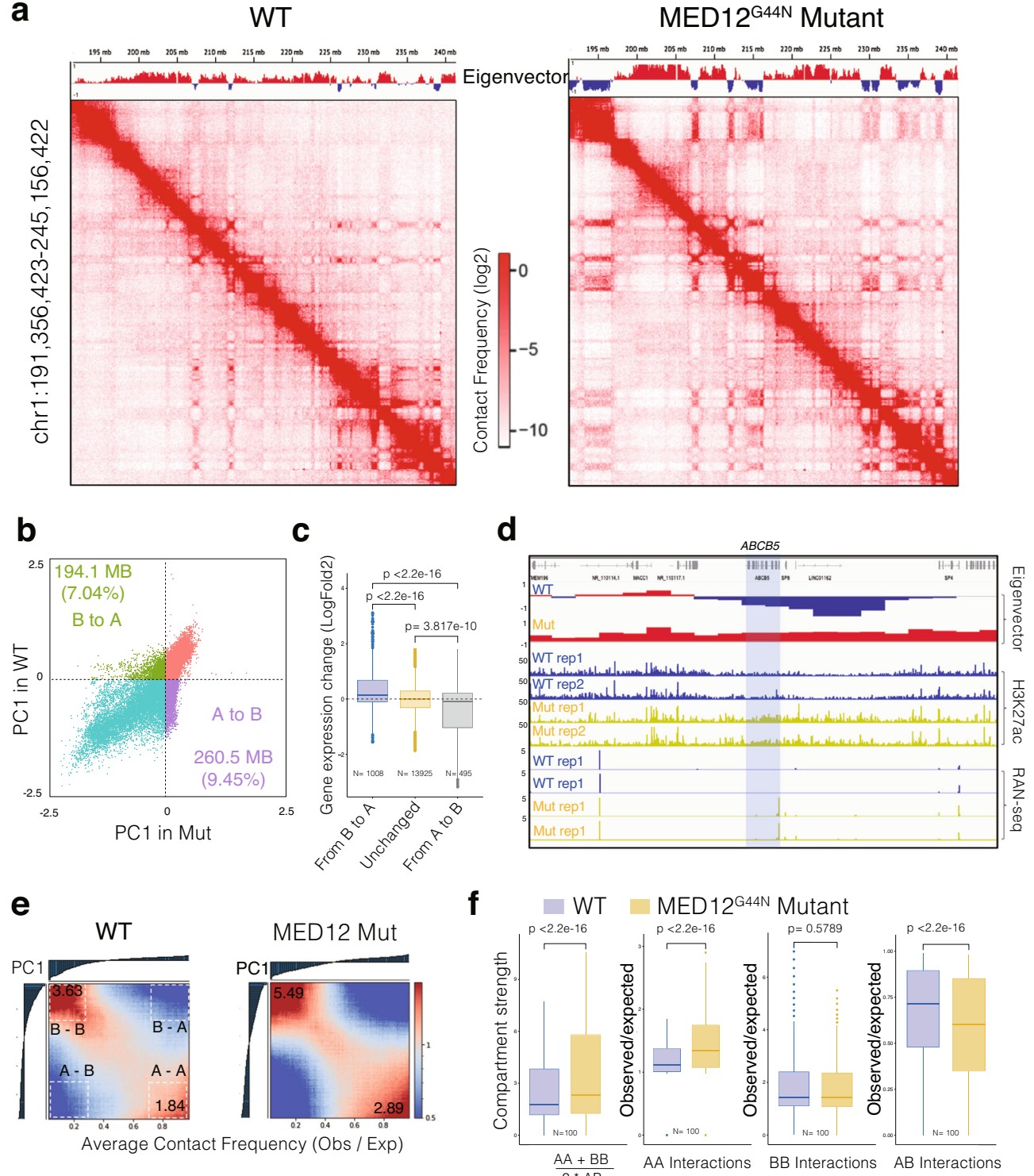

**Fig. 5 | MED12 Gly-44 mutations lead to genome-wide A/B compartment reorganization. a** Hi-C contact frequency maps at 100 kb resolution (chr1: 191 M–245 M) in WT (left) and MED12 Mut (right) cells. The first principal component values (PC1) of the genomic regions are shown at the top. **b** Scatterplot of PC1 in WT versus PC1 in Mut illustrating 194,100,000 bp B to A compartment switch regions and 2605000000 bp A to B compartment switch regions after the MED12 mutation. **c** The boxplots show the log2 fold gene expression change (MED12 mutant vs. WT cells) for the genes located within B to A switch, stable (*n* = 1008), and A to B switch region (*n* = 495). In the boxplot, the center line represents the median, the box contains the interquartile range, and the whiskers extend to the 5th and 95th percentiles. The statistical test was performed using a two-sided

Wilcoxon test (ns = not significant). **d** A regional example displaying the gene ABCB5 is located in the B to A switch region after the MED12 mutation, along with regional H3K27ac binding enrichment and expression fold change increase. **e** Saddle plot displays the genome-wide compartment interaction in wild-type (Left) and MED12 Mut (Right) cells based on Hi-C compartment eigenvectors. **f** The boxplots show the quantification of the compartment strength, A to A, B to B, and A to B interaction strength in WT and Mut cells, (*n* = 100). The center line represents the median, the box contains the interquartile range, and the whiskers extend to the 5th and the 95th percentiles. The statistical test was performed using a two-sided Wilcoxon test (ns = not significant).

system to deeply characterize metabolomics and genomics of pure populations of *MED12* mutant cells. In this study, we utilized CRISPR genome engineering to introduce fibroid-relevant recurrent *MED12* mutation at the endogenous *MED12* locus. We generated multiple clonal populations of *MED12* mutant cells that recapitulate in vitro and in vivo features of UFs.

Notably, more than 50% of all recurrent *MED12* exon two mutations are mutating the Glycine aa at the 44[th] codon into at least six others (aa), indicating that altering Gly at this position results in significant structural and functional alterations in MED12. Interestingly, Gly is the smallest and the only aa that contains Hydrogen as its side chain, whereas all other amino acids contain Carbon[54]. As such, Gly may have a critical impact on both structure and function of the protein because the sidechain-less form of Glycine provides conformational flexibility and can also bind to phosphate, e.g., ATP, in the case of kinase pockets[54]. Our findings also support the clinical[4,55,56] and mouse genetic[32] data that the recurrent MED12 Gly-44 mutations are gain of function and "oncogene" in UFs tumorigenesis[26]. However, the exact mechanism of how these genetic alterations drive UFs pathogenesis is poorly understood. Notably, cell sorting analysis of UFs indicates a presence of a stem-like population that harbors *MED12* mutations and can drive in vivo growth of leiomyoma xenograft tumors[57,58]. The organoid models could be generated from the stem-like cells that are positive for *Stro-1/CD44* markers[59]. Hence, it is plausible to hypothesize that the initial *MED12* mutation occurs in these stem and progenitor cells, which subsequently interact with the surrounding myometrial tissue to give rise to a fibroid tumor. Notably, in this study, we introduced the *MED12* mutation in a terminally differentiated myometrial smooth muscle cell line. Despite that, we observed remarkable disease-relevant cellular and molecular features, indicating the robustness of the MED12 Gly-44 mutation in transforming the cells.

At the biochemical level, a series of elegant experiments indicated that recurrent *MED12* mutations alter mediator complex activity. Specifically, through mass spectrometry analysis in insect cells overexpressing UFs-linked *MED12* mutations, Turunen et al. show that *MED12* mutations disrupt mediator-associated CDK activity[20]. For example, the same group demonstrated that Gly-44 mutant MED12 disrupts allosteric activation of cyclin C-CDK8/19 complex[19]. Notably, the fact that UF-associated genetic alterations in *MED12* are not loss function mutations that cause protein loss but a recurrent mutation that frequently changes the same (Gly-44) amino acid suggests that this mutation may not function as a "loss" of function. Supporting this, we observe that the altered gene expression program in *MED12 Gly-44* mutant cells differs from the *MED12* knock-out cells. These findings suggest that the *MED12 Gly-44* mutations in UFs are not simple "loss of function" in *MED12*.

Our findings also shed some light on *MED12* mutant cells that could not be previously generated. Supporting previous findings[52,53], we observe that the *MED12* mutant cells have lower fitness in 2D culture conditions; however, in 3D sphere conditions, the mutant cells formed larger spheres. Notable, the mutant cells have been cultured in 2D conditions for months period, therefore, it remains to be seen whether these cells will eventually adopt the 2D conditions and gain better fitness in the long run. Significantly, in 3D, we observed enhanced proliferation and expression of collagen genes as seen in primary fibroid tumors[2,27,32]. Our in vivo results from the xenograft kidney capsule model further corroborated these findings and showed that *MED12* mutant cells have higher proliferative capacity and ECM and collagen deposition levels. The results from co-culture and condition experiments suggest that the *MED12* mutant cells have elevated fitness in 3D conditions, and they enhance the proliferation/survival capacity of neighboring non-mutant cells as well. The molecular mechanisms downstream of MED12 mutations that underlie this increased fitness are yet to be identified. One prominent mechanism is the increased expression of collagen genes that remodels the ECM.

This, together with the growth factors released in the environment, results in autocrine and paracrine signaling that improves the overall fitness of mutant and non-mutant cells.

Notably, our findings also indicate a potential therapeutically exploitable vulnerability in the mutant cells as these cells are significantly more sensitive to a chemotherapeutic agent, Carboplatin. Our detailed metabolomics and transcriptomics analysis show that the engineered *MED12* mutation results in robust metabolic and gene expression changes highly reminiscent of primary fibroid tissues. Therefore, our model system may be a valuable asset for the larger research community to study and target UFs-relevant MED12 mutations. Importantly, in addition to being mutated in >70% of UFs[4], *MED12* exon two mutations are also observed in breast fibroadenomas and phyllodes tumors (59%)[60,61], uterine leiomyosarcomas (7–30%)[62], chronic lymphocytic leukemias (5%)[63], and colorectal cancers (0.5%)[62]. Therefore, it is plausible to speculate that the mutant *MED12* in these diseases also results in similarly altered metabolomics, 3D chromatin compartmentalization, and therapeutic vulnerabilities. To this end, the genome-engineered strategy we present here could be explored to introduce similar mutations in other cellular models and comparatively study the outcomes to assess whether the same *MED12* mutations result in similar molecular and cellular phenotypes in these different cell types.

Our study has some limitations. Firstly, we engineered one of the several *MED12* exon two mutations, and whether other recurrent hot spot mutations result in the same phenotype is yet to be determined. Secondly, we introduced the MED12 Gly-44 mutation in an immortalized myometrial smooth muscle cell line. Although this model retains specific markers of smooth muscle/myometrium cell markers observed in primary cultures of SMC[30]; cell lines are generally a poor model that cannot capture the heterogeneous nature of complex tissues. To this end, combining precise CRISPR engineering with UF-organoid models[59] should further advance our understanding of *MED12* mutations and identify MED12-mutation-specific therapeutically exploitable vulnerabilities.

## Methods

### Ethical statement

This research complies with all relevant ethical regulations including human cell line and animal studies. The study protocol was approved by the Ethics Committee of Northwestern University. In vivo experimental procedures were approved by Northwestern University's Animal Care and Use Committee approved all procedures involving animals in this study protocol #IS00012470)and were performed according to respective institutional regulations.

### Cell lines and culture conditions

The human myometrial smooth muscle cell line myo-hTERT Cells were kindly provided by Dr. Jian-Jun Wei (Northwestern University) and are described by Carney et al.[64]. The cell line is listed in the Cellosaurus database under the accession number CVCL_9Z20. These cells and HUtSMC (ATCC, PCS-460-011) were maintained in DMEM/F-12 (Gibco, Invitrogen 11320033) with 10% Fetal Bovine Serum (Fisher scientific, SH3091003) and 1% Penicillin–streptomycin (Life Technologies,15140–122). Cells were cultured and incubated at 37 °C in a humidified atmosphere of 5% $CO_2$ and 95% air.

### MED12(G44N) CRISPR knock-in and nucleofection

MED12-Gly-44 targeted sgRNA oligos sequences (Supplementary Data 4), were designed using CRISPOR software[65] selecting the lentiguide-puro protocol and were ordered from IDT. 10 μM from each oligonucleotide pair were mixed using annealing buffer (10 mM Tris, pH=8, 50 mM NaCl, 1 mM EDTA) in a total volume of 50 μl and incubated at 95 °C for 5 min. Then, they were allowed to slowly cool down to RT. Next, annealed oligos were diluted (1:200) using sterile water.

The ligation reaction was performed using BsmB1-digested 50 ng backbone (Modified P2A_mCherry CROPseq-Guide-puro (# 86708, Addgene), 1 μl of the diluted oligos, and 1XT4 DNA ligase and incubated overnight at 16 °C. The next day, 2.5 μl of the ligation reaction was transformed into NEB Stable competent E.coli (C3040H, NEB) and incubated overnight in the presence of ampicillin selection. Next, several colonies were picked and grown overnight. The next day, plasmid DNA was isolated using a Qiaprep spin miniprep kit (27206, Qiagen) and sent to Sanger sequencing for validation of successful insertion. Modified GFP-hCas9 (Plasmid 41815, Addgene) and ssDNA-HDR template (IDT) (Supplementary Data 4) were used as well for nucleofection.

In all, $4 \times 10^5$ hTERT cells were nucleofected with MED12-sgRNA (1.5 μg), Cas9-GFP (1.5 μg) and single strand HDR template (50pmol) using Neon transfection system (Invitrogen, MPK5000) and Neon 10 μl kit (Thermo Fisher, MPK1025). The parameters used for nucleofection were 1400 Voltage/ 20 with/ 2 pulse. 48 hr later, double positive (mcherry+GFP) cells were selected using FACS and seeded as single cells in 96-well plates.

## Single-cell colony qPCR scanning and validation

Single-cell colonies were split into replicate plates after the colonies grew. One of the replicate plates was washed with PBS twice, Tris-HCL (pH 8.5) was added, and the cells were scraped off with pipette tips. Then, cells were transferred to qPCR plates (4306737, Applied biosystem). Cells were incubated at 95 °C/15 min for lysis, then cooled on ice for 1 min. Then they were treated with Proteinase K (55 °C/30 min) (EO0492, Thermo Fisher) and incubated at 95 °C/10 min for proteinase inactivation. After this, they were transferred to new qPCR plates, and the same amount of lysis was used per reaction (reactions were performed as duplicates). qPCR was performed using Fast SYBR™ Green Master Mix (4385616, Thermo Fisher) and the same reverse primer, with two different forward primers WT/Mut (Supplementary Data 4) to detect positive colonies.

After detecting mutation-positive colonies, the gDNA was extracted from the replicate plates using the Purelink genomic DNA mini kit (K182002, Thermo Fisher), and RNA was extracted using the Zymo research Quick RNA miniprep kit(R1054). Isolated RNA was converted to cDNA (4387406, Applied biosystem). The MED12-exon2 region was amplified using primers F/R (Supplementary Data 4) from gDNA or using F/R primers from cDNA (Supplementary Data 4), then sent for Sanger sequencing for mutation validation.

## CRISPR TIDER analysis

After sorting for double positive (mCherry+GFP) cells, half of the cells were seeded as a population separately from single-cell colonies. These cells were then used to detect population-level CRISPR knock-in rate using CRISPR-TIDER[66] analysis. They were subsequently passaged over 9 weeks to determine if *MED12* mutation abundance changed in the population over time. Also, mutant positive colonies were analyzed using CRISPR-TIDER to differentiate homozygous/heterozygous mutation. For TIDER, three PCR amplicons were produced following the website's protocol (http://shinyapps.datacurators.nl/tider/). Control and sample PCR amplicons were produced using F/R primers on genomic DNA. Reference PCR amplicons were produced using two overlapping primers and the same set of F/R primers for the control and sample PCRs described by the website's protocol (Supplementary Data 4).

Then, sanger sequencing results (ACGT/NU core) (.ab1 files) were uploaded using the default settings on the website.

## Western blotting

Cells were lysed using 1X RIPA buffer, and protein concentrations were determined using the BCA assay (23225, Thermo Fisher). 1 μg/μl protein was mixed with 4X sample buffer with reducing agent and boiled at 95 °C for 10 min. Next, 20 μg of boiled protein was loaded onto either a NuPAGE 4–12%, Bis-Tris gradient gel (# NP0335, Thermo Fisher) or 3–8%, Tris-acetate gel (EA0375, Thermo Fisher), and samples were run at 130 V for about 1.5 h. Proteins were transferred to nitrocellulose membrane using iBlot dry transfer system (Program 3/8 min). Next, membranes were blocked using 5% milk dissolved in TBS-T (20 mM Tris, 150 mM NaCl, 0.1% Tween 20; pH 7.6) for 1 h, rocking at room temperature (RT). After blocking, membranes were incubated with primary antibodies (1:1000 dilution) MED12 (Bethyl lab, #A300-774A), TDO2 (Protein-tech,15880-1AP), Phospho-Histone H2A.X (Ser139) (Cell Signaling, 2577), Anti-β-Actin antibody- clone AC-15, monoclonal (Sigma, Mouse monoclonal, A1978-100μl) prepared in blocking buffer overnight at 4 °C. The next day, membranes were washed with TBS-T 3 times for 5 min. Then, they were incubated with secondary antibodies (1:10,000) (Anti-Rabbit IgG (H + L) (Promega, W4011), Anti-Mouse IgG (H + L) (Promega, W402B) diluted in blocking buffer for 1 h at RT. After the incubation, membranes were again washed 3 times for 10 min. Lastly, membranes were covered with western blot detection reagents (37074, Thermo Fisher) and visualized using the iBright imaging system.

## RNA seq and differential expression/GSEA analysis

Total RNA was isolated using the Zymo research Quick RNA miniprep kit utilizing the on-column DNAse treatment according to the manufacturer's instructions (R1054). The overall RNA purity was assessed by absorbance at 260 and 280 and the potential degradation was assessed by running on the agarose gel. Samples with (A260/A280 ratio ~2 and 28S and 18S band intensity ratio greater than 2 were accepted as pure and non-degraded and processed for qPCR and RNA-Seq analysis. RNA was prepared for sequencing using the NEBNext® Poly(A) mRNA Magnetic Isolation Module (NEBNext, E7490) and NEBNext® Ultra™ Directional RNA Library Prep Kit (NEBNext, E7420) according to the manufacturer's instructions. Paired-end sequencing of all RNA libraries was performed on the Illumina NextSeq 500 Platform.

The quality of FastQ files of RNA seq was checked using FastQC (www.bioinformatics.babraham.ac.uk). RNA-seq reads were aligned to the GRCh38 human genome assembly (Ensembl release 102) using the STAR aligner (v2.7.5)[67] with default settings. BAM files were converted into bigwig files using bam coverage/DeepTools (v3.5.1)[68] (bin size 1, normalized BPM). Gene exon counts were found using featureCounts (Subread package, v1.6.1, settings: -g gene_id -t exon -p -s 2)[69]. Differential expression analysis was performed using the R package DESeq2 (v1.36.0)[70], and the Wald test was used for significance (*p* value). The Independent hypothesis weighting (IHW) method was applied for multiple testing corrections, with the false discovery rate controlled for at (*p* < 0.05). Differentially expressed genes in the MED12(G44N) cells (*p*-adj<0.01) and DE genes in Leiomyoma cells (*p*-adj<0.01) were used for GO term analysis which was performed using the NIH's DAVID Bioinformatics Resource[71] (Supplementary Fig. 6). DESeq2 normalized exon counts were used in GSEA (v4.0.2)[72] analysis in default settings.

All leiomyoma and myometrium RNA-seq data were downloaded from the NCBI-GEO data repository via accession GSE128242 and original publication data processing steps were followed[36]. Heatmaps of differentially expressed genes were plotted in R (cran.r-project.org) using *pheatmap*[73].

## Incucyte live cell imaging

Incucyte Live cell imaging system (Sartorius) was used for tracking cell proliferation. The system took a photo of cell plates every 2 h in different image channels (Phase/Green or Red). For cell nucleus counting, 1 μM SiR-DNA nuclear dye was used (Cytoskeleton, SC007) and captured using the red channel. At the end of the experiment, proliferation data were analyzed using the Incucyte analysis tool and p-values were calculated using the Incucyte raw data. Relative proliferation was normalized to the starting time.

## EdU staining

Cells ($4 \times 10^5$) were seeded in six-well plates one day before Edu staining. Then, cells were treated with 10 μM EdU for 90 min following the manufacturer's protocol (Click-iT™ Plus EdU Alexa Fluor™ 488 Flow Cytometry Assay Kit, Thermo Fisher, C10632). Next, cells were stained for DNA content using FxCycle™ Violet Stain (Thermo Fisher, F10347) (1 μl violet for 1 ml media). Lastly, cells were analyzed using flow cytometry, and the results were analyzed using FlowJo.

## H3K27Ac CUT&Tag and analysis

Benchtop CUT&Tag[41] V.2 protocol was slightly modified to profile genome-wide H3K27ac. Briefly, Concanavalin A-coated (ConA) beads (10 μl/sample) were washed using binding buffer (100 μl/sample) twice and kept on ice until the cells were ready. Cells were then harvested and counted to obtain 100,000 cells for each sample. Then, cells were centrifuged and washed one-time using a wash buffer. After washing, cells were centrifuged and resuspended in wash buffer (100 μl/sample). Next, ConA beads were added to each sample while vortexing gently. The bead-sample mixture was rotated for 10 min at RT. Next, samples were put on a magnet stand to clear the liquid. Following this, samples were resuspended in ice-cold antibody buffer (50 μl/reaction) while vortexing and then kept on ice. Afterward, 3 μl H3K27ac antibody (ab4729, Abcam) was added to each sample while vortexing gently. Samples were incubated overnight at 4 °C on a nutator. The next day, samples were cleared using a magnet stand. A secondary antibody (ABIN101961) mixture (2 μl antibody diluted in 100 μl dig-wash buffer for each sample) was added to each sample while vortexing. After 1 h of incubation at RT on a nutator, samples were cleared and washed twice using Dig-wash buffer (1 ml/sample). Then, the pA-Tn5 adapter complex (2.5 μl pA-Tn5 in 47.5 μl Dig-300 buffer) was added to each sample while vortexing and samples were incubated for 1.5 h at RT on a nutator. Here, we used pAG-Tn5 from EpiCypher (Cat No: 15–1117). After the incubation, samples were cleared and washed twice using Dig-300 buffer (1 ml/sample). Next, 300 μl tagmentatiton buffer was added to each sample, and they were incubated for 1.5 h at 37 °C. To stop tagmentatiton, 10 μl 0.5 M EDTA, 3 μl 10%SDS, and 2.5 μl 20 mg/ml Proteinase K was added to each sample. After adding Proteinase K, samples were vortexed immediately and incubated overnight at 37 °C. The following day, samples were incubated at 50 °C for 30 min. To isolate DNA, 300 μl phenol-chloroform was added to samples and mixed by vortexing. Each mixture was transferred into a phase-lock tube (129046, Qiagen) and centrifuged at $16,000 \times g$ for 3 min at RT. Next, 300 μl chloroform was added to each sample and inverted 10 times to mix. Samples were then centrifuged at $16,000 \times g$ for 3 min at RT. After centrifugation, the aqueous layer was transferred to new tubes containing 750 μl 100% ethanol and mixed well with pipetting. Samples were incubated on ice for 5 min and centrifuged for 15 min at 4 °C $16,000 \times g$. The liquid was removed (the pellet may not be visible), and 1 ml of 100% ethanol was added to rinse the pellet. Then, samples were centrifuged for 1 min at 4 °C $16,000 \times g$. The liquid was carefully removed, and samples were air-dried for about 15 min. Next, the pellet was dissolved using 25 μl TE buffer (10 mM Tris-HCl pH 8, 1 mM EDTA supplemented with 1:400 diluted RNAse A). To remove potential RNA contaminants, samples were then incubated for 10 min at 37 °C. Library preparation was performed as described in Benchtop CUT&Tag V.2 protocol. The library was sequenced using NextSeq 500 (2 × 75 bp) to obtain 5 million reads per sample.

Reads were aligned to the hg38 reference genome using Bowtie2[74]. Then, PCR duplicates and blacklisted regions were removed using Picard tools ("Picard Toolkit." 2019; Broad Institute, GitHub Repository, https://broadinstitute.github.io/picard/) and bedtools, respectively. Bigwig files were generated using deepTools[75] to visualize Cut&Tag tracks. Peak calling was performed using MACS2[76]. To determine differentially acetylated regions, we used DiffBind (Stark R and Brown G (2011)/Bioconductor)[77] package available on R studio.

## Apoptosis assay

Cells were seeded into 96-well plates at a density of $1.5 \times 10^3$ cells/well. The following day, treatments were performed using 65 μM/75 μm Carboplatin (IC30/IC40 concentration, (Selleckchem, # S1215) mixed with 1:1000 diluted Caspase 3/7 dye (10403, Biotum). Then, cells were monitored using the Incucyte live cell imaging system using phase and green channels. The apoptosis rate was determined using the green integrated intensity/confluency values, and the results were plotted using the Incucyte cell imaging analysis.

## Immunofluorescence staining

Approximately $1.5 \times 10^5$ cells were seeded onto coverslips in 6-well plates. The next day, cells were treated with the indicated concentrations of Carboplatin for three days. Then, cells were washed with PBS and fixed using 4% paraformaldehyde in PBS for 10 min at RT. After fixation, cells were washed three times with ice-cold PBS and incubated with 0.25% Triton X-100 in PBS for permeabilization. Next, cells were washed three times for 5 min with PBS and blocked using 1% BSA, 22.52 g/ml glycine in PBS-T (PBS + 0.1 Tween 20) for 1 h at RT. After blocking, cells were incubated with primary antibodies (Phospho-Histone H2A.X (Ser139) (1:1000) (Cell Signaling, # 2577) prepared with 1% BSA in PBS-T overnight at 4 °C in a humidified chamber. The next day, cells were washed three times for 5 min with PBS-T, and then they were incubated with a secondary antibody (Alexa Fluor 594, Invitrogen #A-11012) prepared in 1% BSA in PBS-T for 1 h at RT. Afterward, cells were washed three times for 5 min using PBS-T. Next, coverslips were mounted onto microscopy slides using a mounting medium with DAPI (S36939, Thermo Fisher). Finally, slides were visualized using the EVOS cell imaging system, and the images were analyzed using ImageJ software. Relative γH2AX levels were drawn, and the p-value (Two-sided unpaired t-test) was calculated in Prism-GraphPad (9.4.1).

## Liquid chromatography-mass spectrometry (LC-MS) and analysis

Cells were seeded on 10 cm plates and the medium was completely aspirated once they reached ~70–80% confluence. Then, cells were rinsed with ice-cold PBS twice and added 1 ml 80% (vol/vol) methanol (cooled −80 °C). Then, cells were scraped on ice with a cell scraper and collected lysate in a conical tube. The lysate was incubated at −80 °C for 5 min and vortexed at room temperature for 1 min. Repeat that step two times. Then, the lysate was incubated at −80 °C overnight for protein precipitation. The next day, they were centrifuged at $20,000 \times g$ for 15 min at 4 °C and transferred the supernatant that contained metabolites to a new 1.5 ml conical tube. Then, the pellet was fully dissolved in 8 M urea buffer, and protein concentrations were determined using a BCA assay (Thermo Fisher, #23225). The same amount of metabolomes based on protein amount were submitted for LC-MS. The metabolome was analyzed in the NU metabolomics core facility. Then, all metabolome peak areas of samples were normalized to their TIC (total ion count) and normalized peak area per metabolite results were analyzed on Metaboanalyst 5.0[78] (https://www.metaboanalyst.ca/).

## High-throughput chromosome conformation capture (Hi-C) and data processing

The Hi-C was performed using the Arima-HiC Kit (A510008, Arima Genomics) as instructed by the manufacturer. Approximate 1 million WT/Mut cells were harvested, counted, and fixed with 1% formaldehyde and quenched with 0.125 M glycine at room temperature. The fixed cells were digested with the restriction enzyme and end-labeled with Biotin-14-dATP, followed by proximity ligation. Then reverse-crosslinking was performed to the ligated samples and sheared into 300–500 bp fragments. The Biotin-labeled DNA fragments were then end-repaired following adapter ligation and PCR amplification. Hi-C libraries were generated using KAPA Library

Quantification Kit (KAPA Biosystems) and quality-checked according to the manufacturer's protocol.

The mapping, filtering, and binning of the data were done using the runHiC (v0.8.6) pipeline (https://doi.org/10.5281/zenodo.55324). First, the adapters of the Hi-C FASTQ files were trimmed using Trim Galore (v0.4.5), and then runHiC aligned the trimmed FASTQ files to the hg38 human reference genome with Burrows–Wheeler Aligner. Then, low-quality reads and PCR duplicates were removed. Read pairs were then used to couple aligned reads, and redundant PCR artifacts and read pairs aligned to the same restriction fragments were filtered out before the next stage. The binning stage binned the reads at 5-kb, 10-kb, 50-kb, 1-Mb, 10-Mb, and 50-Mb resolution and performed the ICE normalization at the same time. After the binning state, ICE normalized matrices.mcool files were generated for downstream analyses.

## High-throughput chromosome conformation capture (Hi-C) compartment analysis

The compartment analysis for WT/Mut HiC was performed using cooltools (DOI:10.1101/2022.10.31.514564). The A/B compartments PC1 values at 100-kb resolution were called using *cooltools eigs-cis* command. The scatter plot of PC1 values before and after the MED12 Gly-44 mutation was plotted using the ggplot2 package of R (v4.1.3). The regions with positive PC1 values were identified as compartments A. The regions with negative PC1 values were identified as compartments B. The compartment strength, AA interactions, BB interactions and AB interactions were calculated and visualized using boxplots with Wilcoxon signed-rank test for statistical analysis. The compartment strength was visualized using the saddle plot function implemented in cooltools.

## 3D spheroid formation and co-culture experiment

Cells were seeded (3000 cells per well) on Corning ultra-low attachment plate, 96 wells (# 4515), and spheroid photos were taken using the EVOS (Thermo Fisher Cat# AMF5000) cell imaging system. Then, spheroid volumes were calculated using ImageJ. Results were plotted using Prism software. For the co-culture spheroid experiment, WT, MED12(G44N) mutant, and MED12(KO) hTERT cells were transduced with lentivirus-included mCherry plasmid. For viral production, HEK293T cells were seeded into a 10 cm plate 1 day before transfection. 1 μg pMD2.G (Addgene, Plasmid # 12259), 2 μg psPAX2 (Addgene, Plasmid #12260) and 4 μg of the Modified P2A_mCherry CROP-seq-Guide-puro (# 86708, Addgene) plasmid were co-transfected into HEK293T cells using PEI. Media was refreshed 12 h after transfection. The virus was collected 24 and 48 h after the first media refreshment and filtered through a 0.45 mm filter. For viral transduction, cells were incubated with virus solution diluted in media and supplemented with 0.01 mg/ml polybrene for 24 h. After transduction, the mCherry-positive cells were sorted by flow cytometry.

For the co-culture spheroid experiment, the same number of HUtSMC (ATCC, #PCS-460-011) and mCherry positive hTERT cells (WT/Mut/KO) were seeded together in Corning ultra-low attachment plates (1500/1500 cells per well). After 4 days, spheroid photos were taken using the EVOS cell imaging system and spheroid volumes were calculated using ImageJ. Then, spheroids were collected, and the cells were dispersed using trypsin-EDTA (0.25%) (Gibco, # 25200056). Then the mCherry positive cell rate in the population was calculated using flow cytometry.

## The analysis of tissue microarrays for MED12 leiomyoma

Ten leiomyomas with *MED12* mutations (G44) were selected. *MED12* mutations were detected by Sanger sequencing and confirmed by RT-PCR. Matched myometrium was used for all leiomyoma cases. After histologic evaluation, tissue microarray (TMA) was prepared by a semiautomatic TMA instrument in paraffin-embedded tissues with 1.5 mm in diameter tissue core from regions of leiomyoma and matched myometrium. TMA was sectioned in 4 μm and subjected to hematoxylin and eosin (H&E) and trichrome stains. The stained slides were reviewed by pathologist and then scanned by Hamamatsu scanner for digital image analysis.

The comparison of γH2AX staining ratios between the tissue microarray samples was carried out using Machine Learning. The samples were cropped with identical settings using QuPath Software[79]. The images were then segmented through the Trainable WEKA Segmentation[80] plug-in of Fiji Software[81]. The γH2AX (Phospho-Histone H2A.X (Ser139) (20E3) Rabbit mAb (Cell signaling, # 9718) stained and γH2AX non-stained nucleus shapes were aligned to the plug-in. Thus, classified images were generated following the alignments. The colored pixel distribution of each classified image was obtained through the Image Color Summarizer and the pixel counts of green shapes (indicating unstained nuclei) and red shapes (indicating γH2AX stained nuclei) were compared.

To understand the γH2AX stained nuclei distribution between stromal and myometrial tissues within each sample, a semi-automated analysis was used. Following manual separation of the stromal and myometrial cell types, the nuclei were highlighted on different gradients according to their staining, through Photoshop's Magic Wand tool. The highlighted particles were then evaluated using Fiji software, giving the unstained nucleus counts and γH2AX stained nucleus counts on different tissues of each sample.

## H3K27Ac CUT & Tag analysis

Differential binding analysis was performed using the R package Diffbind (Stark R and Brown G (2011)/Bioconductor)[77]. Enriched and depleted peaks were chosen using a threshold of <0.1 FDR. Heatmaps were generated using Deeptools'[75] computeMatrix and plotHeatmap functions. Peaks were annotated using the R package ChipSeeker[82,83]. Genes were determined to be peak-adjacent if the TSS or TES were within 10 kb of the peak center. RNA-counts were normalized with the R package DESeq2[70] using variance stabilizing transformation (VST). Violin plots of normalized RNA expression were generated using the R package ggplot2 (DOI:10.1007/978-0-387-98141-3) and *p* values were calculated using Wilcoxon signed-rank tests.

## Mouse xenograft experiments

Northwestern University's Animal Care and Use Committee approved all procedures involving animals in this study. Since the method involves hormone treatments, we use adult females at 9–10 weeks of age. $10^6$ live cells (mutant *MED12* and WT) were suspended into 10 μl rat-tail collagen (type I) solution (BD Bioscience, San Jose, CA) and cultured overnight as floating cultures. The next day, cell pellets were grafted underneath kidney capsules of ovariectomized adult female non-obese diabetic-SCID (IL2Rγ null) mice hosts (Jackson Laboratory, Bar Harbor, ME) supplemented with subcutaneous implantation of 50 mg progesterone plus 50 μg estradiol 60-day slow-release pellets (Innovative Research of America Inc., Sarasota, Fl) as described previously[84]. After 8 weeks, the mice were sacrificed. The images of the regenerated tissues on the kidney surface were taken using a dissecting microscope connected to a computer with Leica Application Suite version 3.8 software (Leica Microsystems). The regenerated tissues were harvested, fixed in Davison, and paraffin-embedded. The H&E staining and Masons' trichrome staining were performed by the Northwestern University Histology and Phenotyping Laboratory. The surface area results of the injected cells were obtained through the Freehand selection tool of the FIJI Software and by the Measure command. The surface areas were calculated on the same magnification and the same sections per each sample.

## Masson's Trichrome staining, staining intensity analysis

The intensity measurements were done by setting the sliced images on a saturation level containing only three channels, with the red color

indicating the Trichrome staining. Then these classified images were analyzed on FIJI software by selecting the injected cells. Thus, the mean Red Intensity density from the designated areas was obtained.

## Statistics and reproducibility

Statistical analyses were performed using GraphPad Prism version (9.4.1) Descriptive statistics are reported as means ± standard deviations (SDs). Between-group differences were assessed using Student's *t* tests. (Hi-C) Compartment Analysis were assessed using two-sided Wilcoxon test, and *p* values were generated by log-rank test. Results were considered significant at *p* < 0.05. For in vitro experiments, no statistical method was used to predetermine sample size and no data were excluded from the analyses. For the in vivo experiment power analysis was performed to determine the sample size to achieve 80% power with 50% difference in lesion size. The mice were randomized in terms of injection of WT or mutant cell into left or right kidney capsule. The investigators were not blinded to allocation during experiments and outcome assessment.

## Reporting summary

Further information on research design is available in the Nature Portfolio Reporting Summary linked to this article.

## Data availability

All genomics data produced in this manuscript are available at Gene Expression Omnibus (GEO) under the accession number: GSE226017. Source data are provided with this paper.

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

## Acknowledgements

We thank all members of Adli lab for their critical insights and recommendations during this study, which was supported by a pilot project award (PI: Adli) from Northwestern Uterine Leiomyoma Research Center

(P50HD098580, PI: Bulun). We thank Professor Debu Chakravarti for his critical suggestion on the manuscript.

## Author contributions

M.A. conceptualized the study and wrote the manuscript. K.B. performed the majority of the experiments and performed the data analysis. X.C. performed the Hi-C data analysis. F.A., H.O., F.S.-P., A.J.D., and H.E. helped with experiments and data analysis. Y.F. and J.J.W. helped with the pathological classification and IHC of the TMA of fibroid tumors. F.Y., P.Y., and S.E.B. helped with the overall concepts and conclusions.

## Competing interests

The authors declare no competing interests.
