## [Peer Review File · Nature Communications]

Reviewers' Comments:

Reviewer #1:

Remarks to the Author:

Review NCOMMS-23-0462

In this manuscript, Buyukcelebi et al. have undertaken to first establish and then phenotype at the biological and molecular levels a cell-based model for MED12 mutation-positive uterine fibroids, the predominant subtype among these benign, yet clinically significant tumors. Mutations in the RNA Polymerase II transcriptional Mediator subunit MED12 account for ~70% of uterine fibroids, but the lack of a validated cell-based model has heretofore hampered efforts to better understand the molecular basis of MED12 in disease onset and progression. For reasons presently unclear, primary cells from MED12-mutant uterine fibroid tumors are rapidly lost from 2D cultures, seriously limiting the application of cellular, molecular, and biochemical profiling analyses in a genetically tractable model system necessary to clarify the role of mutant MED12 disease etiology.

To circumvent this barrier, the authors employed a CRISPR-assisted homology-directed gene editing approach to introduce into immortalized human uterine smooth muscle cells (UtSMCs) a targeted mutation at MED12 codon 44, the most frequently altered codon in uterine fibroids. Subsequent biological and molecular phenotyping revealed that the engineered MED12-mutant (MED12 G44N) UtSMCs recapitulated specific features of MED12 mutation-positive uterine fibroid cells and tumors, including reduced and enhanced proliferation in 2D and 3D cultures, respectively, as well as overlapping transcriptomic and metabolic profiles. Notably, the authors show that MED12-mutant UtSMCs accrue higher DNA damage loads and exhibit enhanced sensitivity to platinum-based DNA damaging agents compared to parental UtSMCs. Finally, and among the most new and novel findings, the authors find that mutant MED12 triggers a genome-wide compartmentalization switch in higher-order chromatin structure that correlates with MED12 mutation-induced gene expression changes, revealing novel insight concerning the basis by which mutant MED12 transcriptionally reprograms UtSMCs to a pathologic state.

This is a comprehensive, compelling, straightforward, and for the most part, rigorously executed study that should be of wide interest beyond the immediate gynecological health and disease research community given the broad impact of Mediator in transcriptional control and the fact that MED12 exon 2 mutations are linked to human pathologies in addition to uterine fibroids, including benign breast disease as well as blood and colorectal cancers. The extensive molecular characterization of genetically engineered UtSMC clones generated herein provide novel insight concerning the impact of pathogenic MED12 mutations on higher-order chromatin architecture and gene expression and further provide a valuable new model for the broader scientific research community. Nonetheless, several key missing experiments along with some additional clarifications would better serve to validate the cell model, justify the author's conclusions, and render the manuscript accurately aligned with existing data.

First, as the only established biochemical defect linked to uterine fibroid driver mutations in MED12, the authors should confirm that Mediator-associated CDK8/19 kinase activity is disrupted in their genetically engineered clonal lines, especially since the G44N mutation is not one naturally found in MED12 mutation-positive uterine fibroids. In this regard, the mechanistic basis by which MED12 activates CDK8 (and likely CDK19) occurs through stabilization of the CDK8 activation (T)-loop (see PMID: 33523904). All known uterine fibroid driver mutations, including missense mutations and indels, alter the region in MED12 responsible for CDK8 T-loop stabilization. Further, certain phenotypes observed in MED12-mutant tumors and MED12-mutant primary cells, including R-loop induced replication stress, S-phase cell cycle delay, and impaired growth in 2D culture are recapitulated by chemical inhibition of Mediator kinase activity. Accordingly, the pathogenic changes triggered by UF driver mutations in MED12 are likely mediated, at least in part, by Mediator kinase disruption. The manuscript would be strengthened considerably by confirmation that the MED12 G44N knock-in mutation engineered herein does in fact disrupt Mediator kinase activity. This could be achieved by indirectly monitoring the phosphorylation status of an established Mediator kinase substrate (for example STAT1 S727) or more directly by assessing Mediator-associated kinase activity in an IP/kinase assay.

Second, data purporting to show that MED12-mutant UtSMCs can stimulate non-autonomous cell proliferation is presently unconvincing. First, there is no data to indicate that HUtSMCs do not undergo cell death during 4 day co-cultures. Although perhaps this is unlikely, their percentage in 4-day co-cultures with WT immortalized UtSMCs is less than 15% despite initial seeding at 50%. While this can be explained by slower growth of HUtSMCs compared to immortalized UtSMCs, it remains formerly possible that that MED12-mutant UtSMCs cells are inhibiting death as opposed to promoting proliferation of HUtSMCs. Second, a higher percentage of HUtSMCs in mixed mutant versus WT MED12 spheres means reduced numbers of mutant compared to WT MED12 cells in these spheres, indicating overall reduced cell autonomous proliferation of MED12 mutant cells. This is in contrast to data in Figure 1G showing enhanced proliferation of MED12-mutant cells in 3D spheres. A more direct measure of cell numbers is necessary to justify the author's conclusion regarding the ability of MED12-mutant cells to promote non-autonomous cell proliferation.

Finally, an obvious question to arise from these studies is whether the MED12-mutant UtSMCs cells established herein can form tumors in immunocompromised mice. The authors of the manuscript have pioneered and extensively employed an elegant kidney capsule model for human uterine fibroids. It would therefore be of great interest to determine if the hallmark pathologic phenotypes exhibited by the engineered MED12-mutant UtSMCs, including enhanced growth and ECM production in 3D models, extend to overt uterine fibroid tumor formation *in vivo*. While suggested, this is not considered an essential experiment.

Additional minor concerns and clarifications that should be addressed include the following:

1. Lines 67-72, Paragraph 2 of the Introduction section: Regarding UF driver mutations, recent work has identified mutations in subunits of the histone loading SRCAP complex to mark a distinct uterine fibroid subclass. In fact, SRCAP mutations are more frequent than are FH mutations.
2. Lines 84-86, Paragraph 3 of the Introduction section: Recent studies have clarified the mechanistic basis for MED12-mediated CDK8/19 activation. This is due to direct T-loop stabilization as opposed to earlier allosteric activation models proposed before structural information was available. In this regard, the authors might want to consider adding the following reference: Li et al. (2021) *Sci Adv* 7(3): eabd4484. PMID: 33523904
3. Figure 2F: Steady state levels of the mutant MED12 protein appear to be significantly higher than those of WT MED12 in contrast to Figure 1D, in which mutant and WT MED12 protein levels are more nearly equivalent. What is the explanation for this? Are the levels of mutant MED12 protein stable? Do levels change over time?
4. Figure 3A: Please indicate fold-cut-off, if any, used for differential gene expression.
5. Figure 3C: GSEA FDR values appear high. Are these numbers accurate?
6. Figure S7: What is the Venn diagram on the left under 3D culture conditions? These numbers are not described in the manuscript text.
7. Line 308-310, Paragraph 1, Results Section "The MED1 Gly44 mutation alters DNA synthesis and renders cells sensitive to DNA damaging agents": The statement that "MED12 mutant UF contains the highest number of structural variations in the genome" is incorrect. In fact, MED12-mutant UFs are comparably more genetically stable than are MED12 WT tumors. Further, MED12-mutant tumors are under-represented among UFs carrying complex chromosomal rearrangements characteristic of chromothripsis. See PMID: 34521855; 23738515; 27889101.
8. There needs to be a better discussion of the role of stem vs differentiated uterine smooth muscle cells as the cell of origin for uterine fibroids. The authors of the paper were the first to identify the former as the putative cell of origin for cell transformation and uterine fibroid formation. They identified uterine fibroid stem cells bearing MED12 mutations and showed that these were capable of inciting tumor formation. The manuscript would be better served by a discussion of how the findings herein relate to the stem cell model? For example, early in the Introduction, the authors indicate that "UF tumors are believed to originate mono-clonally from a single mutated smooth muscle cell (SMC)" with no reference to stem cells. Are the authors proposing in this manuscript that uterine fibroid tumors arise from differentiated smooth muscle cells that acquire a MED12 mutation? A deeper discussion of the findings herein and how they may relate to the prevailing model for uterine fibroid tumor formation is warranted in order to better frame the author's new and novel findings within the larger picture of uterine fibroid disease etiology.

Reviewer #2:

Remarks to the Author:

Buyukcelebi et al. used CRISPR knock-in to generate genetically engineered myometrial cells with mutated MED12 gene as a model for uterine myoma tumour. MED12 encodes Mediator complex subunit 12, which regulates transcription initiation and elongation and is mutated in nearly 70% of myometrial tumours. The authors show that this genetic model recapitulates several cellular and molecular features of the fibromas including increased proliferation in 3D cultures, an altered transcriptome with increased expression of DNA repair genes, decreased DNA synthesis rate, an altered epigenome, a genome-wide switch in chromatin compartmentalization and changes in global metabolism with altered tryptophan/kynurein metabolism. Their study shows that the MED12 mutation leads to cell cycle disruption, abnormal DNA replication and repair, making cells susceptible to additional DNA damage from chemotherapeutic agents, such as carboplatin.

Uterine fibroids/uterine myoma is a common non-cancerous gynaecologic condition affecting up to 70% of women and can lead to serious symptoms such as uterine bleeding, anaemia, implantation problems, repeated miscarriages, preterm labour. About 70% of these tumours are driven by somatic mutations in MED12, but mutant MED12 cells cannot be maintained in culture conditions. The genetic model described in this paper has been shown to faithfully represent uterine fibroids. Further characterization of this model will improve understanding of the pathophysiological mechanisms of this disease and thus contribute to the discovery of new therapeutic targets and new treatment options, underscoring the importance of this study.

The results of this study were obtained using state-of-the art techniques, including CRISPR knock-in, whole-genome transcriptomics with RNA sequencing, Cut&Tag to profile histone modifications, high throughput chromosome conformation capture compartment analysis,... These methods were described in detail to allow reproduction.

Major comments

There are some concerns considering the soundness of the experimental data as the authors did not focus sufficiently on several aspects that may contribute to irreproducibility.

Cell lines need to be adequately described with source (no data for hTERT in the manuscript), passage numbers and authentication data.

The number of independent experiments performed have to be indicated. This information is missing for experiments that examined the effects on proliferation in 2D and 3D culture (Figure 1) and effects on the apoptosis (Figure 4). Also, n was not defined for the metabolomics study.

Antibodies have to be adequately described (source, catalogue number, lot number) and validated against a positive and a negative control. Whole membranes should be submitted for peer review. Due to known differences between batches of antibodies validation by the manufacturer is not sufficient. Experiments with anti-TDO2 antibodies and antibodies against phospho-H2AX were performed without the use of a positive and negative control.

The quality of RNA may affect downstream experiments so the RIN number has to be provided (line 829).

The importance of Gly44 for the observed gain of function needs to be supported and explained from a protein structure /MED12 ortholog perspective.

The authors performed nontargeted metabolomics to examine changes in metabolites in MED12 mutated and WT cells and found significant differences with downregulation of 14 metabolites and upregulation of 21 metabolites. They explain that tryptophan was significantly depleted and kynurein elevated, that tryptophan metabolism was the most enriched metabolic pathway, and that tryptophan 2,3-dioxygenase-2 was present only in MED12 mutated cells, demonstrating its involvement in enhanced tryptophan metabolism. This particular metabolic alteration has been

previously found in fibroids, confirming the suitability of this model. However, the authors also need to focus on other, potentially novel findings.

Other comments

Uterine fibroids are also known as uterine myoma, leiomyomas, myoma uteri - this information has to be provided.

The authors have to comment on the recently reported uterine fibroid stem cell-derived organoid model doi: 10.1007/s43032-022-00960-9.

Line 257: The authors need to explain to the reader that they studied acetylation of Lysin27 on Histone H3 which is associated with enhanced transcriptional activation.

Line 313: The authors need to explain to the reader that gamma-H2AX (γH2AX) is a phosphorylated form of histone H2AX and can be used as a marker for double-strand breaks.

More details need to be provided about the clinical tissue samples that were included in analyses (Figure 4)

Materials and Methods

5 and 3 primes should be labelled for all sequences.

Reviewer #3:

Remarks to the Author:

The manuscript entitled "Engineered MED12 mutations drive uterine fibroid-like transcriptional and metabolic programs by altering the 3D genome compartmentalization." by K. Byukcelebi and colleagues aims to aims to develop cellular models based on CRISPR knock-in of the Gly44 mutation in exon 2, to deeply characterize MED12 mutations and better understand the molecular pathways downstream of these mutations.

Authors have demonstrated in previous studies the power of CRISPR tool kit for genome editing. Therefore, this work could be a novel research topic for the identification and develop therapeutic targets to inhibit uterine fibroid tumorigenesis.

Specific comments

Introduction

-Line 67: Authors should specify the meaning of having "a relatively low mutational burden", as this is not exactly the same as having few recurrent genetic mutations.

-Lines 98 to 101: Please clarify and properly reference the sentence "The recurrent alteration in patient samples and the data from the engineered mouse model strongly suggest that altering the Glycine at the 44 amino acid leads to a gain of function "oncogene" mutation that drives UF tumorigenesis". The proposed reference does not mention MED12 or LM, and no references to the mouse model or patients mentioned are provided.

-Line 111: Please state which is the "disease-relevant myometrial SMC line".

-Please be consistent with mutation nomenclature throughout the text (Gly44, G44, Gly-44...).

Results

-Line 159-160: How would you explain that Gly44 mutant cells are less proliferative if they are believed to divide uncontrollably to develop UF? This should be commented in the discussion section.

-Lines 168-169: How do you support the sentence "MED12 Gly 44 mutations are causal in inducing higher cell proliferation and, eventually, tumor formation in humans and mice". Please add the references that supports this statement.

-Lines 233-234: How does the detection of a set of differentially expressed genes support "the

overall hypothesis that the UF-associated MED12 mutations are not loss of function, but a gain of function mutations"? Please elaborate on why those genes may be expressed as a result of a gain of function mutation.

-Line 240: Please provide the list of differentially expressed genes with logFC and adjusted P-values.

-Line 325: Please clarify the intended meaning of "normal" myometrium; is it adjacent myometrium to the MED12 UF or myometrium from fibroid-free patients?

-Line 325: Does n=10 mean 10 samples total, 10 arrays or 10 UF and 10 matched myometrium samples?

-Line 372: Is there any specific region for you to highlight the ABCB5 gene and not any other?

Discussion

-Line 392: Do you have any hypothesis as to why MED12 mutations may be less favourable in 2D cultures?

-Please provide further discussion regarding how genes altered by MED12 mutation can be the cause of tumorigenesis.

Materials and methods

-Line 724-723: Please add the reference to the hTERT human myometrial cell line.

-Perhaps it would be useful to create a Supplementary Table including all primers used rather than stating them along the methods.

-Line 843: Please indicate the meaning of the abbreviation "IHW" and reference the method.

-Line 853: Please correct "pheatmaps" to "pheatmap" and reference the package properly (i.e. to the github link).

-Line 909: Please add the proper reference to MACS2.

-Line 960: Please add the proper reference to runHiC.

-Line 972. Please add the proper reference to cooltools.

-Lines 975 and 1030. Please add the proper reference to ggplot2.

Figures

-Figure 2: The figure layout is rather confusing. Please follow alphabetical order to improve readability.

-Figure 3: In line 499, add "(b)" to primary fibroid tumors and highlight "c" in bold style.

-Figure 4c should include pictures showing the relative immunofluorescence of γ -H2AX signal intensity in 75 μ M Carboplatin-treated WT and mutant MED12 cells. Please, also replace μ m by μ M to avoid confusions with micrometers.

-Figure 4: In line 522 there is a "(g)" that I assume is a mistake that should be removed.

Supplementary figures

-Supplementary Figure 1 is referenced in the text as "Supporting Figure 1" (Line 128)

-Supplementary Figure 5 should include the names of differentially expressed genes.

-Supplementary Figure 7: The same p-value should be established for both comparisons. Additionally, you should clarify what the contrast for the p-value is. Is it significantly differentially expressed genes? Shared genes between 3D cultures and primary UF?

-Supplementary Figure 8: Please correct "Edu" to "EdU" in the legend.

-Supplementary Figure 11: Supplementary Figure 11a should be included as Supplementary Table 2, also including the proper legend.

Supplementary tables

-Supplementary Table 1: The provided file does not have any mentions to it being Supplementary Table 1 and lacks a proper legend.

Minor comments:

- Line 117: Please replace "Gly4" by G44.
- Line 123: "aa" should be "amino acids".
- Line 239: Please replace the end of the sentence "as previously reported" and add the proper reference number related to "Moyo et al."
- Line 262: Please correct "H2K27ac" by "H3K27ac"
- Line 349: Please correct "H3K4m3" to "H3K4me3"

Reviewer #4:

Remarks to the Author:

Buyukcelebi et al. present a manuscript in which they analyze clonal cell lines in which clinically relevant MED12 mutations were introduced by CRISPR. The authors show that in 3D culture these mutations promote growth and even stimulate the proliferation of wild-type cells. The authors go on to perform a number of omics methods to characterize these mutants. Metabolomics results show that the cells are depleted for tryptophan, but enriched for kynurenine, which is correlated to an increase in the levels of TDO2. RNAseq shows an increase in the expression levels of MYC and E2F targets and an upregulation of a set of collagen genes. When the mutant cells are challenged by with carboplatin (a DNA damaging agent) they are more sensitive than the wild-type and show higher levels of apoptosis. Finally, the authors show that in the mutant cells there is a substantial difference in compartmentalization of the genome.

This is an interesting and important study that warrants publication. However, I do have to major comments that need to be resolved.

Regarding the role of MED12 or the Mediator complex the authors should re-examine the existing literature. They claim that the "mediator complex is a critical player in genome organization that links distal regulatory elements to gene promoters". However, one of the papers they cite for this (Jaeger et al), explicitly states that Mediator is not involved in bridging promoters and enhancers. El Khattabi et al. (2019) came to the same conclusion. Therefore it would be good to revise this, so that the text is consistent with the current ideas about Mediator and the role in 3D genome organization. In the same regard, the author should discuss their results in the light of a recent paper in which Hi-C was performed in MED12 knock-out cells (Haarhuis et al. 2022). In this paper compartmentalization was also increased.

One thing that was unclear is to this reviewer it seemed that compartmentalization of the B compartment was increased (example plot Figure 5a). These results are on the face of it very similar to the results of Haarhuis et al. 2022. However, the systematic analysis suggests that the opposite is the case. Considering previous results, it would be good to re-examine these results. Or discuss potential differences.

A second major comment deals with the claims regarding cause and effect. The authors claim that "induced gene expression changes is[sic], in part, due to reprogrammed epigenome, at least of the H3K27ac chromatin states" and that "the differential gene expression changes downstream of MED12 mutation are partly due to altered 3D genome organization". It should be noted that these claims cannot be made, rather only correlations are observed. Although it would make sense that H3K27ac increase lead to a change in expression, the order has not been established here. For the compartment changes, it seems more likely that the compartment switches are actually a consequence of the change in gene expression. When explicit claims with regard to cause or consequence cannot be made the authors should refrain from this.

Minor comments:

- The authors detected ~5500 differentially expressed genes between normal myometrium and UF samples. However, they do not discuss whether there is a significant overlap between the genes found between the mutant clones and the wild-type.
- Use of the word "significant(ly)" should be reserved for when the results of statistical test are discussed.
- The upregulation of MYC and E2F target genes is rather trivial, because the mutant cells are more proliferative.
- "> 90 % of the genes (2124/2359) between WT and mutant cells were consistent across the

mutant clones" I assume differential genes are meant here?

- "the KO cells did form any spheres (not shown)" It is assume that the cells "did not form any spheres".
- "we found 166 upregulated and 181 downregulated genes common between MED12 mutant cells vs fibroid tumors. However, these numbers increased to 539 and 569 genes" These numbers are meaningless if we do not know the expected (background) overlap. Please provide these numbers.
- "the highest number of structural variations in the genome". Highest of what? Of the UF samples? Please state this explicitly.

Point by point Response to reviewers' comment

Reviewer #1 (Remarks to the Author):

In this manuscript, Buyukcebebi et al. have undertaken to first establish and then phenotype at the biological and molecular levels a cell-based model for MED12 mutation-positive uterine fibroids, the predominant subtype among these benign, yet clinically significant tumors. Mutations in the RNA Polymerase II transcriptional Mediator subunit MED12 account for ~70% of uterine fibroids, but the lack of a validated cell-based model has heretofore hampered efforts to better understand the molecular basis of MED12 in disease onset and progression. For reasons presently unclear, primary cells from MED12-mutant uterine fibroid tumors are rapidly lost from 2D cultures, seriously limiting the application of cellular, molecular, and biochemical profiling analyses in a genetically tractable model system necessary to clarify the role of mutant MED12 disease etiology.

To circumvent this barrier, the authors employed a CRISPR-assisted homology-directed gene editing approach to introduce into immortalized human uterine smooth muscle cells (UtSMCs) a targeted mutation at MED12 codon 44, the most frequently altered codon in uterine fibroids. Subsequent biological and molecular phenotyping revealed that the engineered MED12-mutant (MED12 G44N) UtSMCs recapitulated specific features of MED12 mutation-positive uterine fibroid cells and tumors, including reduced and enhanced proliferation in 2D and 3D cultures, respectively, as well as overlapping transcriptomic and metabolic profiles. Notably, the authors show that MED12-mutant UtSMCs accrue higher DNA damage loads and exhibit enhanced sensitivity to platinum-based DNA damaging agents compared to parental UtSMCs. Finally, and among the most new and novel findings, the authors find that mutant MED12 triggers a genome-wide compartmentalization switch in higher-order chromatin structure that correlates with MED12 mutation-induced gene expression changes, revealing novel insight concerning the basis by which mutant MED12 transcriptionally reprograms UtSMCs to a pathologic state.

This is a comprehensive, compelling, straightforward, and for the most part, rigorously executed study that should be of wide interest beyond the immediate gynecological health and disease research community given the broad impact of Mediator in transcriptional control and the fact that MED12 exon 2 mutations are linked to human pathologies in addition to uterine fibroids, including benign breast disease as well as blood and colorectal cancers. The extensive molecular characterization of genetically engineered UtSMC clones generated herein provide novel insight concerning the impact of pathogenic MED12 mutations on higher-order chromatin architecture and gene expression and further provide a valuable new model for the broader scientific research community.

We appreciate these positive and nice comments.

Nonetheless, several key missing experiments along with some additional clarifications would better serve to validate the cell model, justify the author's conclusions, and render the manuscript accurately aligned with existing data.

First, as the only established biochemical defect linked to uterine fibroid driver mutations in MED12, the authors should confirm that Mediator-associated CDK8/19 kinase activity is disrupted in their genetically engineered clonal lines, especially since the G44N mutation is not one naturally found in MED12 mutation-positive uterine fibroids. In this regard, the mechanistic basis by which MED12 activates CDK8 (and likely CDK19) occurs through stabilization of the CDK8 activation (T)-loop (see PMID: 33523904). All known uterine fibroid driver mutations, including missense mutations and indels, alter the region in MED12 responsible for CDK8 T-loop stabilization. Further, certain phenotypes observed in MED12-mutant tumors and MED12-mutant primary cells, including R-loop induced replication stress, S-phase cell cycle delay, and impaired growth in 2D culture are recapitulated by chemical inhibition of Mediator kinase activity. Accordingly, the pathogenic changes triggered by UF driver mutations in MED12 are likely mediated, at least in part, by Mediator kinase disruption. The manuscript would be strengthened considerably by confirmation that the MED12 G44N knock-in mutation engineered herein does in fact disrupt Mediator kinase activity. This could be achieved by indirectly monitoring the phosphorylation status of an established Mediator kinase substrate (for example STAT1 S727) or more directly by assessing Mediator-associated kinase activity in an IP/kinase assay.

Answer: I appreciate the constructive recommendation. We agree with the reviewer that several established reports suggested altered CDK8 kinase activity due to MED12 mutations in leiomyoma. Indeed, this is the topic we are currently actively investigating and is part of a new manuscript in preparation and is yet to be completed. Our experimental data is in line with the reviewers' suggestion and indicate that MED12 mutation indeed disrupts the mediator kinase activity.

Here, we are providing an experimental data (shown in the figure on the right) that MED12 G44 mutant cells have significantly less STAT1 (S727) phosphorylation. We are respectively showing this data for the reviewers' eye only because this data, together with other phosphorylation defect data related to altered Mediator kinase activity, is part of another manuscript that is under preparation.

Second, data purporting to show that MED12-mutant UtSMCs can stimulate non-autonomous cell proliferation is presently unconvincing. First, there is no data to indicate that HUtSMCs do not undergo cell death during 4-day co-cultures. Although perhaps this is unlikely, their percentage in 4-day co-cultures with WT immortalized UtSMCs is less than 15% despite initial seeding at 50%. While this can be explained by slower growth of HUtSMCs compared to immortalized UtSMCs, it remains formerly possible that that MED12-mutant UtSMCs cells are inhibiting death as opposed to promoting proliferation of HUtSMCs. Second, a higher percentage of HUtSMCs in mixed

mutant versus WT MED12 spheres means reduced numbers of mutant compared to WT MED12 cells in these spheres, indicating overall reduced cell autonomous proliferation of MED12 mutant cells. This is in contrast to data in Figure 1G showing enhanced proliferation of MED12-mutant cells in 3D spheres. A more direct measure of cell numbers is necessary to justify the author's conclusion regarding the ability of MED12-mutant cells to promote non-autonomous cell proliferation.

Answer: We agree with the reviewer that we do not have data that MED12 mutant cells may also prevent cell death of WT cells surrounding mutant cells. This is certainly a possibility, and we now acknowledge this in the main text that this may be due to increased proliferation/survival. Furthermore, we believe that the indicated discrepancy that the cells were mixed 50:50, but the rate of WT cells seemed to be only 15% is due to the fact that this experiment was performed four days of sphere formation. During this time, we think mutant cells are proliferating better than the non-mutant cells.

However, to better understand whether mutant cells directly play role in the proliferation of the neighboring WT cells, we have also performed a new experiment to get better insight into the non-autonomous effect of Med12 mutation. Specifically, we performed a long-term live cell imaging-based cell proliferation assay (Incucyte platform) to assess whether a conditioned media from MED12 mutant cells will have a direct impact on the proliferation of non-mutant cells. The data shown on the right side shows that 20% conditioned media from MED12 mutant cells significantly enhances the proliferation of non-mutant cells and this effect is significantly greater than the effect of condition media from WT cells. This new data is now presented at Supplementary Figure 4c.

Figure: Condition media from MED12 mutant cells enhances proliferation of WT cells.

Finally, an obvious question to arise from these studies is whether the MED12-mutant UtSMCs cells established herein can form tumors in immunocompromised mice. The authors of the manuscript have pioneered and extensively employed an elegant kidney capsule model for human uterine fibroids. It would therefore be of great interest to determine if the hallmark pathologic phenotypes exhibited by the engineered MED12-mutant UtSMCs, including enhanced growth and ECM production in 3D models, extend to overt uterine fibroid tumor formation in vivo. While suggested, this is not considered an essential experiment.

Answer: This is a great suggestion. We did not initially focus on the in vivo phenotypes of these engineered cells because we wanted to highlight that these cells are providing a novel in vitro

model to study the biology of fibroid specific MED12 mutations. However, per reviewers' comment, we performed the xenograft experiment suggested by the reviewer. This new data, which is presented as **Figure 1i** and **Figure 1j** in the main figure demonstrates that the MED12 mutant cells form significantly larger lesions compared to WT cells, which form minimal regenerated tissue at the injected site in the kidney capsule (**Figure i, below**). Furthermore, the lesions/regenerated tissue from the MED12 Gly44 mutant cells deposited/produced significantly higher levels of collagens and ECM, as measured by the Masson's trichrome staining.

We thank the reviewer for this experiment, which we further highlight the utility of our engineered cells. These new in vivo data are now presented in the main text and shown in the main figure as **Figure 1i** and **Figure 1j**.

Additional minor concerns and clarifications that should be addressed include the following:

1. Lines 67-72, Paragraph 2 of the Introduction section: Regarding UF driver mutations, recent

work has identified mutations in subunits of the histone loading SRCAP complex to mark a distinct uterine fibroid subclass. In fact, SRCAP mutations are more frequent than are FH mutations.

Answer: We thanks the reviewer for the valuable comment. We have now revised introduction text and cited the indicated reference.

2. Lines 84-86, Paragraph 3 of the Introduction section: Recent studies have clarified the mechanistic basis for MED12-mediated CDK8/19 activation. This is due to direct T-loop stabilization as opposed to earlier allosteric activation models proposed before structural information was available. In this regard, the authors might want to consider adding the following reference: Li et al. (2021) Sci Adv 7(3): eabd4484. PMID: 33523904

Answer: Thanks for your comment, the corresponding reference has been cited.

3. Figure 2F: Steady state levels of the mutant MED12 protein appear to be significantly higher than those of WT MED12 in contrast to Figure 1D, in which mutant and WT MED12 protein levels are more nearly equivalent. What is the explanation for this? Are the levels of mutant MED12 protein stable? Do levels change over time?

Answer: We have noticed this as well, however, this has not been a consistent phenotype, but we often do see higher levels of MED12 mutant proteins. We think this might indicate that the mutant protein is more stable compared to the WT protein as we do not see significant difference in mRNA production.

4. Figure 3A: Please indicate fold-cut-off, if any, used for differential gene expression.

Answer: We have used Padj values to identify differentially expressed genes.

5. Figure 3C: GSEA FDR values appear high. Are these numbers accurate?

Answer: Yes, they are accurate. We have now updated the figure with actual FDR numbers. In GSEA analyses, the FDR of 0.25 is nearly equal to the classic FDR of 0.05. below is an explanation provided by the GSEA website: https://software.broadinstitute.org/cancer/software/gsea/wiki/index.php/FAQ#Where_are_the_GSEA_statistics_.28ES.2C_NES.2C_FDR.2C_FWER.2C_nominal_p_value.29_described.3F

“Why does GSEA use a false discovery rate (FDR) of 0.25 rather than the more classic 0.05?”

An FDR of 25% indicates that the result is likely to be valid 3 out of 4 times, which is reasonable in the setting of exploratory discovery where one is interested in finding candidate hypothesis to be further validated as a results of future research. Given the lack of coherence in most expression datasets and the relatively small number of gene sets being analyzed, using a more stringent FDR cutoff may lead you to overlook potentially significant results. For more

information about gene set enrichment analysis results, see Interpreting GSEA in the GSEA User Guide.”

6. Figure S7: What is the Venn diagram on the left under 3D culture conditions? These numbers are not described in the manuscript text.

Answer: We have now updated the text to better describe these numbers and the Figure legend of this Supplementary figure has been expanded to better clarify the figure.

7. Line 308-310, Paragraph 1, Results Section “The MED1 Gly44 mutation alters DNA synthesis and renders cells sensitive to DNA damaging agents”: The statement that “MED12 mutant UF contains the highest number of structural variations in the genome” is incorrect. In fact, MED12-mutant UFs are comparably more genetically stable than are MED12 WT tumors. Further, MED12-mutant tumors are under-represented among UFs carrying complex chromosomal rearrangements characteristic of chromothripsis. See PMID: 34521855; 23738515; 27889101.

Answer: Thanks for your valuable comment. We have revised the text and the indicated references are now properly cited. Upon careful reading of the indicated literature, we have revised the indicated section in the main text and no longer emphasize that the MED12 mutant UFs have highest levels of structural variations.

8. There needs to be a better discussion of the role of stem vs differentiated uterine smooth muscle cells as the cell of origin for uterine fibroids. The authors of the paper were the first to identify the former as the putative cell of origin for cell transformation and uterine fibroid formation. They identified uterine fibroid stem cells bearing MED12 mutations and showed that these were capable of inciting tumor formation. The manuscript would be better served by a discussion of how the findings herein relate to the stem cell model? For example, early in the Introduction, the authors indicate that “UF tumors are believed to originate mono-clonally from a single mutated smooth muscle cell (SMC)” with no reference to stem cells. Are the authors proposing in this manuscript that uterine fibroid tumors arise from differentiated smooth muscle cells that acquire a MED12 mutation? A deeper discussion of the findings herein and how they may relate to the prevailing model for uterine fibroid tumor formation is warranted in order to better frame the author’s new and novel findings within the larger picture of uterine fibroid disease etiology.

Answer: we appreciate this valuable comment. We have now discussed this point in the discussion section of the manuscript.

Reviewer #2 (Remarks to the Author):

Buyukcelebi et al. used CRISPR knock-in to generate genetically engineered myometrial cells with mutated MED12 gene as a model for uterine myoma tumour. MED12 encodes Mediator complex subunit 12, which regulates transcription initiation and elongation and is mutated in nearly 70% of myometrial tumours. The authors show that this genetic model recapitulates several cellular and molecular features of the fibromas including increased proliferation in 3D cultures, an altered transcriptome with increased expression of DNA repair genes, decreased DNA synthesis rate, an altered epigenome, a genome-wide switch in chromatin compartmentalization and changes in global metabolism with altered tryptophan/kynurein metabolism. Their study shows that the MED12 mutation leads to cell cycle disruption, abnormal DNA replication and repair, making cells susceptible to additional DNA damage from chemotherapeutic agents, such as carboplatin.

Uterine fibroids/uterine myoma is a common non-cancerous gynaecologic condition affecting up to 70% of women and can lead to serious symptoms such as uterine bleeding, anaemia, implantation problems, repeated miscarriages, preterm labour. About 70% of these tumours are driven by somatic mutations in MED12, but mutant MED12 cells cannot be maintained in culture conditions. The genetic model described in this paper has been shown to faithfully represent uterine fibroids. Further characterization of this model will improve understanding of the pathophysiological mechanisms of this disease and thus contribute to the discovery of new therapeutic targets and new treatment options, underscoring the importance of this study.

The results of this study were obtained using state-of-the art techniques, including CRISPR knock-in, whole-genome transcriptomics with RNA sequencing, Cut&Tag to profile histone modifications, high throughput chromosome conformation capture compartment analysis,... These methods were described in detail to allow reproduction.

We thank the reviewer, and we appreciate such kind and encouraging comments

Major comments

There are some concerns considering the soundness of the experimental data as the authors did not focus sufficiently on several aspects that may contribute to irreproducibility.

Cell lines need to be adequately described with source (no data for hTERT in the manuscript), passage numbers and authentication data.

Answer: We thank the reviewer for this constructive comment. We have now revised the methods section where we added the indicated information. The human myometrial cell line myo-hTERT Cells were kindly provided by Jian-Jun Wei (Northwestern University) and it was provided from C. Mendelson (UT Southwestern). These sources have been properly cited now.

The number of independent experiments performed have to be indicated. This information is

missing for experiments that examined the effects on proliferation in 2D and 3D culture (Figure 1) and effects on the apoptosis (Figure 4). Also, n was not defined for the metabolomics study.

Answer: We thank the reviewer for the comment and the number of independent experiments was added to the figure where applicable. All experiments were performed in triplicate. The 2D data is a triplicate well data from one representative experiment. And the 3D sphere formation data is from multiple spheres from one of the biological triplicates.

Antibodies have to be adequately described (source, catalogue number, lot number) and validated against a positive and a negative control. Whole membranes should be submitted for peer review. Due to known differences between batches of antibodies validation by the manufacturer is not sufficient. Experiments with anti-TDO2 antibodies and antibodies against phospho-H2AX were performed without the use of a positive and negative control.

Answer: We thank the reviewer for suggesting these improvements. We have now provided more detail information about each antibody. The full membrane versions of the western blots are now presented as **Supplementary Figure 12**. We did not have positive and negative control for all antibodies. However, since we have genetically manipulated the cells, we believe the best control for MED12 antibody as well as the TDO2 ab is the fact that the KO cells have no expression of MED12 protein as well as TDO2 protein.

The quality of RNA may affect downstream experiments, so the RIN number has to be provided (line 829).

Answer: We thank the reviewer for this valuable comment. The overall RNA purity was assessed by absorbance at 260 and 280 and the potential degradation was assessed by running on agarose gel. Samples with (A260/A280 ratio ~ 2 and 28S and 18S band intensity ratio greater than 2) were accepted as pure and non-degraded and processed for qPCR and RNA-Seq analysis.

The importance of Gly44 for the observed gain of function needs to be supported and explained from a protein structure /MED12 ortholog perspective.

Answer: We thank the reviewer for this valuable comment. Recent structural studies (Li et al. (2021) Sci Adv 7(3), PMID: 33523904) have highlighted the mechanistic basis for MED12-mediated CDK8/19 activation. It was shown that the Gly-44 mutation likely impacts T-loop stabilization as opposed to earlier allosteric activation models proposed before structural information was available. We have also checked the alpha-fold predicted structure of human MED12 protein. As indicated by Li et al, the Gly-44 position is at the turn of the loop that docks on the CDK8.

The authors performed nontargeted metabolomics to examine changes in metabolites in MED12 mutated and WT cells and found significant differences with downregulation of 14 metabolites and upregulation of 21 metabolites. They explain that tryptophan was significantly depleted and kynurein elevated, that tryptophan metabolism was the most enriched metabolic pathway, and that tryptophan 2,3-dioxygenase-2 was present only in MED12 mutated cells, demonstrating its involvement in enhanced tryptophan metabolism. This particular metabolic alteration has been previously found in fibroids, confirming the suitability of this model. However, the authors also need to focus on other, potentially novel findings.

Answer: We agree that these valuable insights. This manuscript is a first in vitro model of fibroid tumors. As such, we aimed to present as much data as possible to highlight that these cells that we engineered are recapitulating well-established transcriptional and metabolic features of fibroid tumors. We have focused on Kynurenine pathway not necessarily because it is a novel pathway that we discovered but rather to demonstrate that the metabolic phenotype of these engineered cells are well in line with the metabolic features of published data (DOI: 10.1038/bjc.2017.361). Although other altered metabolic pathways (such as altered Valine, leucine and isoleucine biosynthesis and Cysteine and methionine metabolism) could be critical for the survival of MED12 mutant cells, we are focusing on such vulnerability pathways in another study, which is in preparation.

Other comments

Uterine fibroids are also known as uterine myoma, leiomyomas, myoma uteri - this information has to be provided.

Answer: We thank the reviewer for the comment and the alternative name of uterine fibroids are added in the beginning of the introduction.

The authors have to comment on the recently reported uterine fibroid stem cell-derived organoid model doi: 10.1007/s43032-022-00960-9.

Answer: Thanks very much for mention this study, the study isolated myometrium stem cells of normal and fibroids from patients and study the organoids. We have now cited this study in our discussion. We believe that such primary organoids could also be a great model and in the future studies, we plan to compare our cells with such organoids. It would be interesting to determine what percent of the organoids from primary patient samples are actually Med12 WT and Mutant cells.

Line 257: The authors need to explain to the reader that they studied acetylation of Lysin27 on Histone H3 which is associated with enhanced transcriptional activation.

Answer: Thanks very much for your comment we respond to your comment, and we added to the explanation at the relevant section.

Line 313: The authors need to explain to the reader that gamma-H2AX (gH2AX) is a phosphorylated form of histone H2AX and can be used as a marker for double-strand breaks.

Answer: Thanks very much for your comment. We have revised the text to convey this message.

More details need to be provided about the clinical tissue samples that were included in analyses (Figure 4)

Answer: We have now provided better description of this TMA resource in **Methods** section.

Materials and Methods

5 and 3 primes should be labelled for all sequences.

Answer: Thank you. We have now revised the text to indicate the direction and we now provide all oligos as a **Table 4**.

Reviewer #3 (Remarks to the Author):

The manuscript entitled “Engineered MED12 mutations drive uterine fibroid-like transcriptional and metabolic programs by altering the 3D genome compartmentalization.” by K. Byukcelebi and colleagues aims to aims to develop cellular models based on CRISPR knock-in of the Gly44 mutation in exon 2, to deeply characterize MED12 mutations and better understand the molecular pathways downstream of these mutations.

Authors have demonstrated in previous studies the power of CRISPR tool kit for genome editing. Therefore, this work could be a novel research topic for the identification and develop therapeutic targets to inhibit uterine fibroid tumorigenesis.

Specific comments

Introduction

-Line 67: Authors should specify the meaning of having “a relatively low mutational burden”, as this is not exactly the same as having few recurrent genetic mutations.

Answer: We thank the reviewer for the valuable comments, we edited indicated the paragraph to highlight that the low rate of mutation is compared to normal cancers.

-Lines 98 to 101: Please clarify and properly reference the sentence “The recurrent alteration in patient samples and the data from the engineered mouse model strongly suggest that altering the Glycine at the 44 amino acid leads to a gain of function “oncogene” mutation that drives UF tumorigenesis”. The proposed reference does not mention MED12 or LM, and no references to the mouse model or patients mentioned are provided.

Answer: We have now provided reference and revised the sentence.

-Line 111: Please state which is the “disease-relevant myometrial SMC line”.

Answer: Thanks for your comment we have revised the text to clarify this

-Please be consistent with mutation nomenclature throughout the text (Gly44, G44, Gly-44...).

Answer: Thank you very much for your comment. We keep consistence with Gly-44 through the manuscript.

Results

-Line 159-160: How would you explain that Gly44 mutant cells are less proliferative if they are believed to divide uncontrollably to develop UF? This should be commented in the discussion section.

Answer: Thank you very much for your valuable comment. Our data show that in 2D culture conditions the MED12 mutant cells are not able to proliferate as much as the WT cells due to, likely a lack of proper ECM structure. However, our data shows in 3D conditions. The MED12 mutant cells are able to express substantially higher collagen and other ECM proteins, which allows them to proliferate faster.

-Lines 168-169: How do you support the sentence “MED12 Gly 44 mutations are causal in inducing higher cell proliferation and, eventually, tumor formation in humans and mice”. Please add the references that supports this statement.

Answer: We have added the proper references

-Lines 233-234: How does the detection of a set of differentially expressed genes support “the overall hypothesis that the UF-associated MED12 mutations are not loss of function, but a gain of function mutations”? Please elaborate on why those genes may be expressed as a result of a gain of function mutation.

Answer: We now updated this sentence to highlight the following conclusion. Our data suggest that the gene expression changes due to MED12 Mutations are nearly completely different than the altered gene expression due to MED12 KO (Supplementary Figure 5). We therefore concluded that MED12 mutations are likely a gain of function mutation, rather than loss of function.

-Line 240: Please provide the list of differentially expressed genes with logFC and adjusted P-values.

Answer: We have now provided this list as **Table 2**.

-Line 325: Please clarify the intended meaning of “normal” myometrium; is it adjacent myometrium to the MED12 UF or myometrium from fibroid-free patients?

Answer: Normal mean adjacent to MED12 UF. We have revised the text to reflect this

-Line 325: Does n=10 mean 10 samples total, 10 arrays or 10 UF and 10 matched myometrium samples?

Answer: n=10 means ten independent patient samples in total.

-Line 372: Is there any specific region for you to highlight the ABCB5 gene and not any other?

Answer: We showed this gene is it was one of the highly differentially regulated gene based on RNA-Seq data.

Discussion

-Line 392: Do you have any hypothesis as to why MED12 mutations may be less favourable in 2D cultures?

Answer: Thanks very much for your comment, we think in 2D culture conditions, the mutant cell can't grow well due to lack of extracellular matrix. In the revised discussion, we tried to convey this message.

-Please provide further discussion regarding how genes altered by MED12 mutation can be the cause of tumorigenesis.

Answer: Thank you for this valuable comment. We have now expanded the discussion section to highlight this.

Materials and methods

-Line 724-723: Please add the reference to the hTERT human myometrial cell line.

Answer: We have revised the material method section and now describe and cite the source of this cell line.

-Perhaps it would be useful to create a Supplementary Table including all primers used rather than stating them along the methods.

Answer: Thank you very much for this advice. We have now included these in the **Table 4**.

-Line 843: Please indicate the meaning of the abbreviation "IHW" and reference the method.

Answer: We Thank the reviewer for the comment, we add the full name of IHW (independent hypothesis weighting) at the applicable section.

-Line 853: Please correct "pheatmaps" to "pheatmap" and reference the package properly (i.e., to the github link).

-Line 909: Please add the proper reference to MACS2.

-Line 960: Please add the proper reference to runHiC.

-Line 972. Please add the proper reference to cooltools.

-Lines 975 and 1030. Please add the proper reference to ggplot2

Answer: We thank the reviewer for the comment, we now fixed the indicated typo sand added the proper references to all the analysis packages.

-Figure 2: The figure layout is rather confusing. Please follow alphabetical order to improve readability.

Answer: We agree with the reviewer, however, despite putting a lot of effort into it, due to the incoherent size of each figure panel, we decided that the current format is the least destructive for the reader.

-Figure 3: In line 499, add “(b)” to primary fibroid tumors and highlight “c” in bold style.

Answer: Thank you. We have now fixed it.

-Figure 4c should include pictures showing the relative immunofluorescence of γ -H2AX signal intensity in 75 μ M Carboplatin-treated WT and mutant MED12 cells. Please, also replace um by μ M to avoid confusions with micrometers.

Answer: We thank the reviewer for the comment. The detailed pictures along with DAPI staining and γ H2AX immunofluorescence for all treatment doses shown in **Supplementary Figure 9**. We appreciate the comment on um vs μ M. We have corrected it.

-Figure 4: In line 522 there is a “(g)” that I assume is a mistake that should be removed.

Answer: Thank you very much for your comment, we fixed it.

Supplementary figures

-Supplementary Figure 1 is referenced in the text as “Supporting Figure 1” (Line 128)

Answer: Thank you very much for your comment, we fixed it.

-Supplementary Figure 5 should include the names of differentially expressed genes.

Answer: We now present the gene names in this figure in **Table 3**

-Supplementary Figure 7: The same p-value should be established for both comparisons. Additionally, you should clarify what the contrast for the p-value is. Is it significantly differentially expressed genes? Shared genes between 3D cultures and primary UF?

Answer: We have considered this suggestion and performed the analysis as suggested. However, there is significant size difference in sample size between two data sets. While the primary patient's sample has data from 15 patient samples while the engineered cells are in duplicate or triplicate. Therefore, under the same P_{adj} conditions of <0.05 , there are ($n= 8135$) differentially expressed genes in primary patient data while the engineered cells have $n=2359$ differentially expressed genes. This ~ 4 fold difference in number is largely technical because of the higher number of primary patient samples, which is considered biological replicate in the analysis. We, therefore, decided to use different P_{adj} values so that we get a comparable number of differentially expressed genes and then perform the overlap analysis.

-Supplementary Figure 8: Please correct “Edu” to “EdU” in the legend.

Answer: Thank you very much for your comment, we fix it.

-Supplementary Figure 11: Supplementary Figure 11a should be included as Supplementary Table 2, also including the proper legend.

Answer: Thank you very much for your comment. We have considered this, However, since this table contained limited information, we think presenting it in the supplementary figure will be more accessible to the reader.

Supplementary tables.

-Supplementary Table 1: The provided file does not have any mentions to it being Supplementary Table 1 and lacks a proper legend.

Answer: We have now provided better legend and cite it in the main text.

Minor comments:

-Line 117: Please replace “Gly4” by G44.

Answer: Thank you very much for your comment, per recommendation from other reviewers, we know consistently call them Gly-44.

-Line 123: “aa” should be “amino acids”.

Answer: Thank you for pointing these errors. We have fixed them.

-Line 239: Please replace the end of the sentence “as previously reported” and add the proper reference number related to “Moyo et al.”

Answer: Thank you very much we fixed the sentence and properly cited in the reference.

-Line 262: Please correct “H2K27ac” by “H3K27ac”

-Line 349: Please correct “H3K4m3” to “H3K4me3.”

Answer: Thank you very much for your comments, we fix this mistake.

Reviewer #4 (Remarks to the Author):

Buyukcelebi et al. present a manuscript in which they analyze clonal cell lines in which clinically relevant MED12 mutations were introduced by CRISPR. The authors show that in 3D culture these mutations promote growth and even stimulate the proliferation of wild-type cells. The authors go on to perform a number of omics methods to characterize these mutants. Metabolomics results show that the cells are depleted for tryptophan, but enriched for kynurenine, which is correlated to an increase in the levels of TDO2. RNAseq shows an increase in the expression levels of MYC and E2F targets and an upregulation of a set of collagen genes. When the mutant cells are challenged by with carboplatin (a DNA damaging agent) they are more sensitive than the wild-type and show higher levels of apoptosis. Finally, the authors show that in the mutant cells there is a substantial difference in compartmentalization of the genome.

This is an interesting and important study that warrants publication. However, I do have to major comments that need to be resolved.

We are thankful for these encouraging words.

Regarding the role of MED12 or the Mediator complex the authors should re-examine the existing literature. They claim that the “mediator complex is a critical player in genome organization that links distal regulatory elements to gene promoters”. However, one of the papers they cite for this (Jaeger et al), explicitly states that Mediator is not involved in bridging promoters and enhancers. El Khattabi et al. (2019) came to the same conclusion. Therefore it would be good to revise this, so that the text is consistent with the current ideas about Mediator and the role in 3D genome organization. In the same regard, the author should discuss their results in the light of a recent paper in which Hi-C was performed in MED12 knock-out cells (Haarhuis et al. 2022). In this paper compartmentalization was also increased.

One thing that was unclear is to this reviewer it seemed that compartmentalization of the B compartment was increased (example plot Figure 5a). These results are on the face of it very similar to the results of Haarhuis et al. 2022. However, the systematic analysis suggests that the opposite is the case. Considering previous results, it would be good to re-examine these results. Or discuss potential differences.

Answer: We deeply appreciate this comment. We have now expanded the discussion section highlight this point. Specifically, we have added the following section to our discussion.

“Interestingly, recent report suggested that MED12/CDK module of mediator complex controls global gene expression program, in part, by limiting the formation of dense heterochromatin domains and 3D chromatin compartmentalization¹⁵. Specifically, Haarhuis et al., observed that MED12 loss enhances H3K9me3-marked heterochromatin domain formations and genome compartmentalization¹⁵. Notably, the enhanced genome compartmentalization that we observe in MED12 Gly-44 mutant cells resemble the findings from MED12 depleted cells.

However, while the loss of MED12 resulted in increased heterochromatin domains and reduced the “open” chromatin regions marked with H3K4me1¹⁵, our results show that MED12 Gly-44 mutations resulted in significantly higher open chromatin regions (marked by H3K27ac, Figure 3e). Furthermore, we find distinct gene expression changes due to MED12 Gly-44 mutation vs loss of MED12 (**Supplementary Figure 5**). Overall, these findings indicate that although Hi-C chromatin compartmentalization in MED12 Gly-44 mutant cells and MED12 KO cells have some resemblance, MED12 Gly44 mutations result in aberrant expression of a distinct set of genes and altered chromatin accessibility compared to WT or MED12 depleted cells.”

A second major comment deals with the claims regarding cause and effect. The authors claim that “induced gene expression changes is[sic], in part, due to reprogrammed epigenome, at least of the H3K27ac chromatin states” and that “the differential gene expression changes downstream of MED12 mutation are partly due to altered 3D genome organization”. It should be noted that these claims cannot be made, rather only correlations are observed. Although it would make sense that H3K27ac increase lead to a change in expression, the order has not been established here. For the compartment changes, it seems more likely that the compartment switches are actually a consequence of the change in gene expression. When explicit claims with regard to cause or consequence cannot be made the authors should refrain from this.

Answer: We deeply appreciate this constructive criticism. We agree with the reviewer regarding the cause and effect. We have revised our text to include this message from the reviewer. We are thankful for this insight.

Minor comments:

- The authors detected ~5500 differentially expressed genes between normal myometrium and UF samples. However, they do not discuss whether there is a significant overlap between the genes found between the mutant clones and the wild-type.

Answer: Our new analysis and results, presented in **Supplementary Figure 7** highlights that there is a significant overlap between differentially regulated genes due to MED12 mutation in engineered cells and primary fibroid samples.

- Use of the word “significant(ly)” should be reserved for when the results of statistical test are discussed.

Answer: We appreciate the comment, we have now revised the text to avoid using it unless we provide the p value and the statistical test.

- The upregulation of MYC and E2F target genes is rather trivial, because the mutant cells are more proliferative.

Answer: We agree with this comment, we highlighted them to partly demonstrate this point.

- “> 90 % of the genes (2124/2359) between WT and mutant cells were consistent across the mutant clones” I assume differential genes are meant here?

Answer: Yes, we meant differentially expressed genes. We have edited the text to highlight this.

- “the KO cells did form any spheres (not shown)” It is assume that the cells “did not form any spheres”.

Answer: We apologize and thankful to the reviewer for highlighting this error. Yes, we meant to say that the cells “did not form any spheres”. We revised the text now.

- “we found 166 upregulated and 181 downregulated genes common between MED12 mutant cells vs fibroid tumors. However, these numbers increased to 539 and 569 genes” These numbers are meaningless if we do not know the expected (background) overlap. Please provide these numbers.

Answer: We thank for this constructive comment. We, thus, performed additional analysis on this issue. We now present and updated **Supplementary Figure 7** where we now also present expected vs observed number of overlapped genes. The new analysis is still in line with overall conclusion that there is significant number of differentially expressed and overlapped genes between the engineered MED12 G44N cells vs. primary UFs. However, the 3D culture conditions is further enhancing the number of common genes for differentially downregulated genes. We therefore, revise the main text to only highlight this.

- “the highest number of structural variations in the genome”. Highest of what? Of the UF samples? Please state this explicitly.

Answer: Thanks for your valuable comment. We have revised the text and the indicated references are now properly cited. Upon careful reading of the indicated literature, we have revised the section in the main text and no longer emphasize that the MED12 mutant UFs have highest levels of structural variations.

Reviewers' Comments:

Reviewer #1:

Remarks to the Author:

In their revised manuscript, Buyukcelebi et al. have responded satisfactorily to this reviewer's initial concerns. The revised manuscript is an improved version, and certainly represents a thorough, compelling, and rigorous study that should be of wide interest beyond the immediate gynecological health and disease research community. The extensive molecular characterization of genetically engineered uterine smooth muscle cell (UtSMC) clones generated herein provide important new insight concerning the impact of pathogenic MED12 mutations on higher-order chromatin architecture and gene expression. In particular, compelling data to support a genome-wide compartmentalization switch in higher-order chromatin structure that correlates with MED12 mutation-induced gene expression changes reveals new insight concerning the basis by which mutant MED12 transcriptionally reprograms UtSMCs to a pathologic state.

The addition of considerable new data in the revised manuscript has not only raised the scientific rigor but also enhanced the biological significance of the study. In this regard, the authors include compelling new data to demonstrate that MED12-mutant, compared to MED12 WT, UtSMCs form tumors in immunocompromised mice. This is an important addition to show that hallmark pathologic phenotypes exhibited by the engineered MED12-mutant UtSMCs, including enhanced growth and ECM production in 3D models in vitro, extend to overt uterine fibroid tumor formation in vivo.

As the first such genetically tractable in vitro model for MED12 mutation-positive uterine fibroids, this study will provide a valuable platform for the broader scientific research community to investigate the genomics of recurrent MED12 mutations and further spur efforts for therapeutic discovery. I therefore support publication of the manuscript in its revised form

Thomas G. Boyer

Reviewer #2:

Remarks to the Author:

Buyukcelebi et al. have responded to most of the comments made by this reviewer. However, authentication data and passage number for hTERT is still missing. The authors provide the following information: "The human myometrial smooth muscle cell line myo hTERT Cells were kindly provided by Dr. Jian-Jun Wei (Northwestern University) and are described by Li et al." In the cited paper, "HMGA2-mediated tumorigenesis through angiogenesis in leiomyoma," myo-hTERT is mentioned only briefly, but no description or reference to characterization of this cell line is given.

In the study by Buyukcelebi et al. genetically modified myometrial (hTERT) cells with mutated MED12 gene are presented as a model for uterine myoma tumours. Therefore, it is important to identify this cell line unambiguously, not only by indicating the tissue of origin and source (where and when the cells were obtained), but also by the correct name (myo-hTERT is not listed in Cellosaurus) and Research Resource Identifier (RRID). Contamination of cell lines shared by different laboratories is common (doi.org/10.1002/ijc.25242), therefore, the authors need to provide information on the authentication of this cell line and also the passage number, as this may increase genetic and phenotypic instability ([doi: 10.1007/978-1-61779-080-5_4](https://doi.org/10.1007/978-1-61779-080-5_4)).

Reviewer #3:

Remarks to the Author:

Reviewer acknowledges the effort made by the authors in assessing the concerns raised during the revision and value their response. They have also conducted additional experiments and make over several figures and language throughout the now revised manuscript. Overall, they have enriched the rigor of the data and communication of the experimental approach, data analyses,

and description of results. Consequently, I recommend to the editor, the publication of this article in Nature Communications.

Reviewer #4:

Remarks to the Author:

The authors have addressed my comments. I support publication of this manuscript.

POINT BY POINT RESPONSE TO REVIEWERS' COMMENTS

Reviewer #1 (Remarks to the Author):

In their revised manuscript, Buyukcelebi et al. have responded satisfactorily to this reviewer's initial concerns. The revised manuscript is an improved version, and certainly represents a thorough, compelling, and rigorous study that should be of wide interest beyond the immediate gynecological health and disease research community. The extensive molecular characterization of genetically engineered uterine smooth muscle cell (UtSMC) clones generated herein provide important new insight concerning the impact of pathogenic MED12 mutations on higher-order chromatin architecture and gene expression. In particular, compelling data to support a genome-wide compartmentalization switch in higher-order chromatin structure that correlates with MED12 mutation-induced gene expression changes reveals new insight concerning the basis by which mutant MED12 transcriptionally reprograms UtSMCs to a pathologic state.

The addition of considerable new data in the revised manuscript has not only raised the scientific rigor but also enhanced the biological significance of the study. In this regard, the authors include compelling new data to demonstrate that MED12-mutant, compared to MED12 WT, UtSMCs form tumors in

immunocompromised mice. This is an important addition to show that hallmark pathologic phenotypes exhibited by the engineered MED12-mutant UtSMCs, including enhanced growth and ECM production in 3D models in vitro, extend to overt uterine fibroid tumor formation in vivo.

As the first such genetically tractable in vitro model for MED12 mutation-positive uterine fibroids, this study will provide a valuable platform for the broader scientific research community to investigate the genomics of recurrent MED12 mutations and further spur efforts for therapeutic discovery. I therefore support publication of the manuscript in its revised form

Thomas G. Boyer

Answer: We thank Dr. Boyer for his time and constructive critiques during the review of this manuscript. We are happy that he is fully satisfied with our improvements.

Reviewer #2 (Remarks to the Author):

Buyukcelebi et al. have responded to most of the comments made by this reviewer.

However, authentication data and passage number for hTERT is still missing. The authors provide the following information: "The human myometrial smooth muscle cell line myo hTERT Cells were kindly provided by Dr. Jian-Jun Wei (Northwestern University) and are described by Li et al." In the cited paper, "HMGA2-mediated tumorigenesis through angiogenesis in leiomyoma," myo-hTERT is mentioned only briefly, but no description or reference to

characterization of this cell line is given.

In the study by Buyukcelebi et al. genetically modified myometrial (hTERT) cells with mutated MED12 gene are presented as a model for uterine myoma tumours. Therefore, it is important to identify this cell line unambiguously, not only by indicating the tissue of origin and source (where and when the cells were obtained), but also by the correct name (myo-hTERT is not listed in Cellosaurus) and Research Resource Identifier (RRID). Contamination of cell lines shared by different laboratories is common (doi.org/10.1002/ijc.25242), therefore, the authors need to provide information on the authentication of this cell line and also the passage number, as this may increase genetic and phenotypic instability (doi: 10.1007/978-1-61779-080-5_4).

Answer: We thank the reviewer, and we fully agree about the importance of cell line authentication. We have updated our materials and methods section to highlight the true source of this cell line that we engineered. The cell line we used for engineering is listed in the Cellosaurus Database under the accession number of CVCL_9Z20.

(https://www.cellosaurus.org/CVCL_9Z20)

We have now updated our materials and methods to better describe this cell line.

Reviewer #3 (Remarks to the Author):

Reviewer acknowledges the effort made by the authors in assessing the

concerns raised during the revision and value their response. They have also conducted additional experiments and make over several figures and language throughout the now revised manuscript. Overall, they have enriched the rigor of the data and communication of the experimental approach, data analyses, and description of results. Consequently, I recommend to the editor, the publication of this article in Nature Communications.

Answer: We thank this reviewer for his time and constructive critiques during the review of this manuscript. We are happy that she/he is fully satisfied with our improvements.

Reviewer #4 (Remarks to the Author):

The authors have addressed my comments. I support publication of this manuscript.

Answer: We thank this reviewer for his time and constructive critiques during the review of this manuscript. We are happy that she/he is fully satisfied with our improvements.